# JNK signaling triggers spermatogonial dedifferentiation during chronic stress to maintain the germline stem cell pool in the *Drosophila* testis

Salvador C Herrera[1], Erika A Bach[1,2]*

[1]New York University School of Medicine, New York, United States; [2]Helen L. and Martin S. Kimmel Center for Stem Cell Biology, New York University School of Medicine, New York, United States

**Abstract** Exhaustion of stem cells is a hallmark of aging. In the *Drosophila* testis, dedifferentiated germline stem cells (GSCs) derived from spermatogonia increase during lifespan, leading to the model that dedifferentiation counteracts the decline of GSCs in aged males. To test this, we blocked dedifferentiation by mis-expressing the differentiation factor *bag of marbles* (*bam*) in spermatogonia while lineage-labeling these cells. Strikingly, blocking *bam*-lineage dedifferentiation under normal conditions in virgin males has no impact on the GSC pool. However, in mated males or challenging conditions, inhibiting *bam*-lineage dedifferentiation markedly reduces the number of GSCs and their ability to proliferate and differentiate. We find that *bam*-lineage derived GSCs have significantly higher proliferation rates than sibling GSCs in the same testis. We determined that Jun N-terminal kinase (JNK) activity is autonomously required for *bam*-lineage dedifferentiation. Overall, we show that dedifferentiation provides a mechanism to maintain the germline and ensure fertility under chronically stressful conditions.
DOI: https://doi.org/10.7554/eLife.36095.001

*For correspondence:
erika.bach@nyu.edu

Competing interests: The authors declare that no competing interests exist.

## Introduction

A robust stem cell pool is critical to the maintenance of highly proliferative tissues during an organism's lifetime. Adult tissue stem cells typically reside in niches, anatomically defined as microenvironments that support their 'stemness' and promote their proliferation, resulting in some daughter cells that retain stem cell characteristics and some that begin differentiation (*Morrison and Spradling, 2008*; *Blanpain et al., 2014*; *Spradling et al., 2011*). Depletion or reduction of stem cells results in tissue atrophy, and the exhaustion of stem cell function is a hallmark of aging (*López-Otín et al., 2013*; *Wang and Jones, 2011*). Thus, the mechanisms that maintain stem cells during an individual's lifespan are of critical importance to understanding the relationship between stem cells and aging and to develop therapies against aging-related clinical conditions like infertility and Parkinson's.

The stem cell pool is dynamic and responds to insults, including injury and starvation in both invertebrate and mammalian model organisms (*Angelo and Van Gilst, 2009*; *Tetteh et al., 2016*; *van Es et al., 2012*; *Yang and Yamashita, 2015*; *McLeod et al., 2010*; *Li and Jasper, 2016*). Recently, dedifferentiation has emerged as a conserved mechanism underlying the replenishment of the stem cell pool after stem cell depletion (*Merrell and Stanger, 2016*). In the mouse intestine, extensive radiation ablates Lgr5-positive crypt stem cells and lineage-tracing experiments revealed that secretory cells dedifferentiate into Lgr5-positive stem cells following this insult (*van Es et al., 2012*). In the *Drosophila* intestine, complete starvation induces the loss of all intestinal stem cells, and polyploid enterocyte cells undergo a reduction in ploidy (called amitosis) and transform into

**eLife digest** From the heart to the brain, our bodies are made of a collection of cells that are specialized to perform precise roles. Yet, certain organs host 'stem cells', which can become any kind of tissue. For example, the testicles of the fruit fly contain germline stem cells; when one of these cells divides, a daughter remains unspecialized, while the other specializes – or differentiates – to become sperm. Despite previous beliefs, a cell that is undergoing specialization can dedifferentiate to become a stem cell again.

As the organism gets older, stem cells become 'exhausted': they divide less, and lose their ability to remain unspecialized. Scientists therefore proposed that dedifferentiation could be a way to replenish a dwindling pool of stem cells, and ward off the effects of age. However, this line of thought has not been tested in the laboratory.

Here, Herrera and Bach tried to test this assumption by creating two populations of male fruit flies. One was genetically intact and the other was modified so that the cells that would become sperm cells could not dedifferentiate to become germline stem cells again. The insects were then raised in either a standard environment (plenty of food and no females) or in stressful conditions (periods of starvation with or without mating).

The experiments showed that dedifferentiation was important to maintain a robust germline stem cell pool, both in the short and long term. This, however, was only the case in the difficult environment; the ability to dedifferentiate made no difference in the easier living conditions. In addition, Herrera and Bach observed that, in the flies' testicles, stem cells obtained through dedifferentiation divided much more often than the 'original' stem cells. Finally, further analyses highlighted a series of genes that are required for dedifferentiation.

That stem cells coming from dedifferentiated cells divide at a higher rate could be relevant to scientists across various fields. For example, this knowledge may help those who study how tissues regenerate after injury, a process that involves dedifferentiation. It also may be used as a cautionary tale for researchers who work on induced pluripotent stem cells – which are created in the laboratory by dedifferentiating specialized cells. This may be especially important because these cells could one day be put in patients to treat diseases such as Parkinson's or Alzheimer's.
DOI: https://doi.org/10.7554/eLife.36095.002

intestinal stem cells (*Lucchetta and Ohlstein, 2017*). In *Drosophila* gonads, after forced differentiation of all germline stem cells (GSCs), differentiating spermatogonia revert to the stem cell state and become functional GSCs (*Brawley and Matunis, 2004*; *Kai and Spradling, 2004*; *Sheng et al., 2009*). While these previous studies showed that dedifferentiation indeed occurs after acute insults or injuries, they did not address its functional significance in these events. Here, we test the functional importance of dedifferentiation through a new genetic approach. We have developed a genetic technique to indelibly mark the cells undergoing dedifferentiation, while at the same time functionally inhibiting the process.

We used the *Drosophila* testis for these studies because of the powerful genetic techniques available in this organism and the broad knowledge about the biology of this organ and its various cell types. In this tissue, approximately 8–14 GSCs reside in a quiescent niche (*Greenspan et al., 2015*). GSCs adhere to niche cells and undergo oriented mitosis, resulting in one daughter cell that retains the stem cell state and remains in contact with the niche (*Figure 1A*). The other GSC daughter cell (the gonialblast) is physically displaced from the niche. After encapsulation by somatic support cells, this latter daughter cell begins differentiation through four rounds of mitotic divisions with incomplete cytokinesis, resulting in 2-, 4-, 8- and 16-cell spermatogonial cysts, the lattermost of which undergoes meiosis to produce 64 spermatids. At the 4- and 8-cell cyst stage, germ cells express *bag of marbles* (*bam*), which is necessary and sufficient for their differentiation (*Sheng et al., 2009*; *Gönczy et al., 1997*). The testis niche also supports a somatic stem cell population called cyst stem cells (CySCs) that produces somatic support cells, which exit the cell cycle and ensheath differentiating GSC daughter cells.

During aging, the population of GSCs declines such that at 50 days of adulthood ~35% of GSCs are lost from the niche and the remaining GSCs have reduced proliferation (*Boyle et al., 2007*;

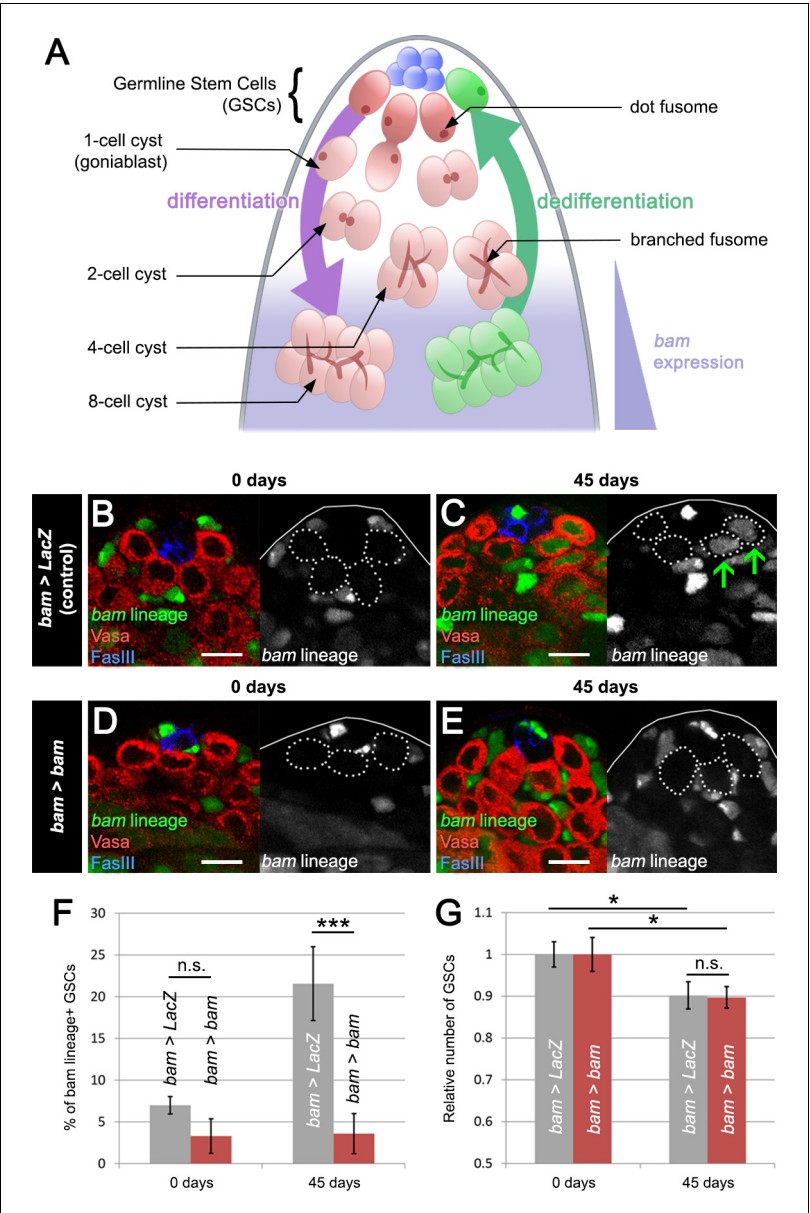

**Figure 1.** Blocking *bam*-lineage dedifferentiation does not protect the GSC pool under normal laboratory conditions. (**A**) Schematic of the *Drosophila* testis. Germline stem cells (GSCs) undergo differentiation through several rounds of mitotic divisions with incomplete cytokinesis (the cells remain connected through a structure, the fusome). During the transition from 4- to 8-cell stage, cells acquire the expression of the differentiation factor *bam*. Germline cells at these stages can undergo dedifferentiation returning to a stem cell state. (**B–E**) Representative images of the testis stem cell niche in control *bam-Gal4, UAS-LacZ* (*bam > LacZ*, **B, C**) or *bam-Gal4, UAS-bam* (*bam > bam,* **D, E**), where we blocked *bam*-lineage dedifferentiation at 0 days and 45 days under normal aging conditions. The *bam* lineage is labeled in green, the germline (Vasa) is red and the niche (FasIII) is blue. Each GSC is outlined by a dashed line. There are no GSCs derived from the *bam* lineage in the control or *bam > bam* testis at 0 days (**B, D**). At day 45, the control contains *bam*-lineage positive GSCs (indicating that they arose through dedifferentiation) (**C**, arrows). However, at the same age, there are no *bam*-lineage positive GSCs in the *bam > bam* testis, demonstrating the efficacy of our technique (**E**). (**F**) Percentage of *bam*-lineage dedifferentiated GSCs in *bam > LacZ* (gray bars) and *bam > bam* (red bars) testes at 0 and 45 days. The proportion of *bam*-lineage GSCs significantly increases in the control *bam > LacZ* at 45 days, while this value remains unchanged in *bam > bam* flies. (**G**) Relative number of GSCs at 0 and 45 days (see Materials and methods for details). In control *bam > LacZ* testes (gray bars), the relative number of GSCs declines from 0 to 45 days. If *bam*-lineage dedifferentiation contributes to the maintenance of a robust GSC pool during aging, we predict that blocking

*Figure 1 continued on next page*

*Figure 1 continued*

dedifferentiation would enhance the decline in total GSC number. However, there is no statistical difference between the control and *bam > bam* genotypes at 45 days. Scale bars represent 10 μm. Bars on charts represent mean ± SE. *p<0.05, ***p<0.001.

DOI: https://doi.org/10.7554/eLife.36095.003

The following figure supplements are available for figure 1:

**Figure supplement 1.** Methodology used for blocking and labeling *bam*-lineage dedifferentiation.
DOI: https://doi.org/10.7554/eLife.36095.004
**Figure supplement 2.** Lineage and real-time expression analysis of *bam-Gal4* in aging and starvation.
DOI: https://doi.org/10.7554/eLife.36095.005

*Wallenfang et al., 2006*). The 35% reduction in the GSC pool in aged males is much smaller than predicted. The average half-life of a GSC is 14 days, and for a testis with 10 GSCs at day 0 of adulthood, there should be <1 GSC at 50 days (*Boyle et al., 2007*; *Wallenfang et al., 2006*). In other words, the reduction in the total GSC pool should be more than 90% at 50 days. This discrepancy in predicted vs observed size of the GSC pool raised the possibility that a mechanism such as spermatogonial dedifferentiation could be responsible for the apparent resistance of the GSC pool to the deleterious effects of aging (*Wang and Jones, 2011*; *Wallenfang et al., 2006*; *Cheng et al., 2008*). However, to date no study has tested this hypothesis by specifically inhibiting dedifferentiation in spermatogonia.

Certain genetic manipulations (transient removal of responses to niche signals or transient misexpression of the key differentiation factor *bam*) cause all GSCs to differentiate. However, upon silencing of these triggers, spermatogonia break apart, migrate back to the niche, outcompete the resident CySCs and become functional GSCs by transducing JAK/STAT signals and repressing *bam* expression (*Brawley and Matunis, 2004*; *Sheng et al., 2009*; *Sheng and Matunis, 2011*). Interestingly, these studies revealed that the 8-cell spermatogonial cyst is the oldest stage still competent to dedifferentiate. *bam*-lineage labeling analysis of 4- and 8-cell spermatogonial cysts revealed that the proportion of dedifferentiated cells in the GSC pool increases with aging; in 50 day old males, ~40% of the GSCs are derived from *bam*-lineage spermatogonia that dedifferentiated (*Cheng et al., 2008*).

Here, we have developed a methodology that enabled us to inhibit specifically dedifferentiation of *bam*-expressing, 4- and 8-cell spermatogonial cysts without apparent side effects, while at the same time lineage-tracing these cells. This has allowed us to address the long-term effects of dedifferentiation. Surprisingly and contrary to predictions, we find that *bam*-lineage dedifferentiation is not required to maintain the GSC pool during aging under normal laboratory conditions. However, it is critical to maintain a robust GSC pool under chronically stressful conditions. Our methodology also allows the identification of dedifferentiated GSCs from their non-dedifferentiated siblings, facilitating the comparison of their characteristics. We find that *bam*-lineage dedifferentiated GSCs have a higher proliferative rate compared to their sibling GSCs in the same testis. Finally, we show that Jun N-terminal kinase (JNK) signaling is activated in germ cells during recovery from stress. By inhibiting JNK activity in *bam*-expressing spermatogonia, we demonstrate that this pathway is essential for dedifferentiation of these cells.

## Results

### Dedifferentiation of *bam*-lineage cells does not protect the GSC pool during aging under standard conditions

In order to study the contribution of dedifferentiation to the maintenance of the GSC pool, we lineage-traced spermatogonial cells. Similar to a previous study (*Cheng et al., 2008*), we used a *bam-Gal4* line, expressed specifically in 4- and 8-cell spermatogonial cysts, to drive *UAS-Flippase (Flp)* expression, and this Flp in turn excises an *FRT-stop-FRT* from the *ubiP63E-FRT-stop-FRT-GFP* cassette (*Evans et al., 2009*). After recombination, GFP becomes an indelible marker of differentiating germ cells that had expressed *bam-Gal4*, and GFP persists even if the cell turns off the *bam* promoter. Since transient germline mis-expression of *bam* is sufficient to induce germline differentiation

(*Sheng et al., 2009*; *Ohlstein and McKearin, 1997*), we speculated that mis-expression of additional Bam protein in these *bam-Gal4*-positive spermatogonia (referred to as *bam > bam*) should prevent them from undergoing dedifferentiation (*Figure 1—figure supplement 1*). As a control and to maintain similar titration of the Gal4 protein, we mis-expressed a neutral construct *UAS-LacZ* by *bam-Gal4* in a second set of flies (referred to as *bam > LacZ*). For reasons unknown to us but also observed by another group (*Cheng et al., 2008*), some somatic support cells are labeled for real-time and lineage *bam* expression (*Figure 1B* and *Figure 1—figure supplement 2C*). We note that this methodology will likely not label all dedifferentiating germ cells, as it has been speculated that gonialblasts and 2-cell spermatogonia can revert to become GSCs. We also note that the efficiency of Flp is not 100%, and so we are not labeling all *bam*-lineage cells. In this study, we refer to GFP-positive germ cells as '*bam*-lineage positive' and 'dedifferentiated', and GFP-negative germ cells as '*bam*-lineage negative' and 'wild type siblings'.

We first analyzed the role of dedifferentiation in control *bam > LacZ* and experimental *bam > bam* males during standard aging conditions, that is, maintaining flies at a low density, in the absence of females and on standard food. In control flies, we observed a significant increase in the percentage of GFP-positive GSCs derived from dedifferentiated *bam*-lineage spermatogonia, from 7.0% in young flies to 21.6% in 45-day-old males (*Figure 1B,C,F*), consistent with a prior study (*Cheng et al., 2008*). As expected, GFP-positive germ cells lose *bam* expression in the process of reverting to a GSC identity (*Figure 1—figure supplement 2A–B''*). However, *bam* mis-expression in the *bam* lineage effectively blocked lineage dedifferentiation, as there was no significant increase in the percentage of lineage-dedifferentiated GSCs in aged males, from 3.3% in young flies to 3.6% in 45-day-old flies (*Figure 1D,E,F*). These results demonstrate that *bam* mis-expression is an effective way to prevent dedifferentiation.

Prior work has shown that the number of GSCs decreases slightly during aging under normal laboratory conditions, and this has led to the model that dedifferentiation provides a means to offset normal GSC loss during lifespan (*Wang and Jones, 2011*; *Wallenfang et al., 2006*; *Cheng et al., 2008*). If this hypothesis is correct, we would expect a further reduction in GSC number after *bam* mis-expression by *bam-Gal4*. Because the starting number of GSCs varies from strain to strain, we compared the relative number of GSCs between *bam > LacZ* and *bam > bam* flies. To our surprise, we found that the relative number of GSCs decreases significantly by 10% after 45 days in both genotypes (from 10.4 to 9.4 cells in *bam > LacZ*, and from 7.9 to 7.1 cells in *bam > bam*) (*Figure 1G* and *Supplementary file 1*). Furthermore, we did not observe any notable differences between aged *bam > LacZ* and aged *bam > bam* testes, as all stages of spermatogenesis appeared similar between the two genotypes (see Figure 4A–B and G–H). We note, however, that due to the fact that some of the *bam*-expressing cells might not undergo recombination (due to incomplete Flp efficiency), we cannot rule out the possibility that unlabeled cells may dedifferentiate and help to maintain the GSC pool under normal laboratory conditions. Additionally, prior work has shown that symmetric renewal, whereby a gonialblast swivels to gain direct access to the niche, occurs at low levels in testes from wild type males (*Sheng and Matunis, 2011*). It is possible that by blocking *bam*-lineage dedifferentiation, we are shifting the equilibrium towards symmetric renewal, but live imaging will be needed to test this hypothesis. Nevertheless, although unexpected and contrary to our predictions, our results strongly suggest that dedifferentiation of the *bam*-lineage does not play an important role in maintaining the GSC pool during aging under normal laboratory conditions.

## *bam*-lineage dedifferentiation accelerates the recovery of the GSC pool after starvation

We speculated that dedifferentiation could be vital during challenging life conditions, such as starvation. Previous work has shown that 6–15 days of protein deprivation (also generally referred to in the literature as 'starvation') causes a 25% reduction in total GSC number (*Yang and Yamashita, 2015*; *McLeod et al., 2010*). Furthermore, the number of GSCs recovered to original levels after 5 days of refeeding on standard food (*McLeod et al., 2010*). We hypothesized that the recovery of GSCs during refeeding could result from dedifferentiating spermatogonia. To test this model, we subjected *bam > LacZ* flies to 15 days of protein starvation, followed by a 9-day time course of refeeding (*Figure 2A*). We assessed the proportion of dedifferentiated GSCs at the end of the starvation period and at different time points during the refeeding phase. Similar to these prior reports, we observed a 33% decrease in the relative number of GSCs after starvation (10.4 at 0 days and 7.0

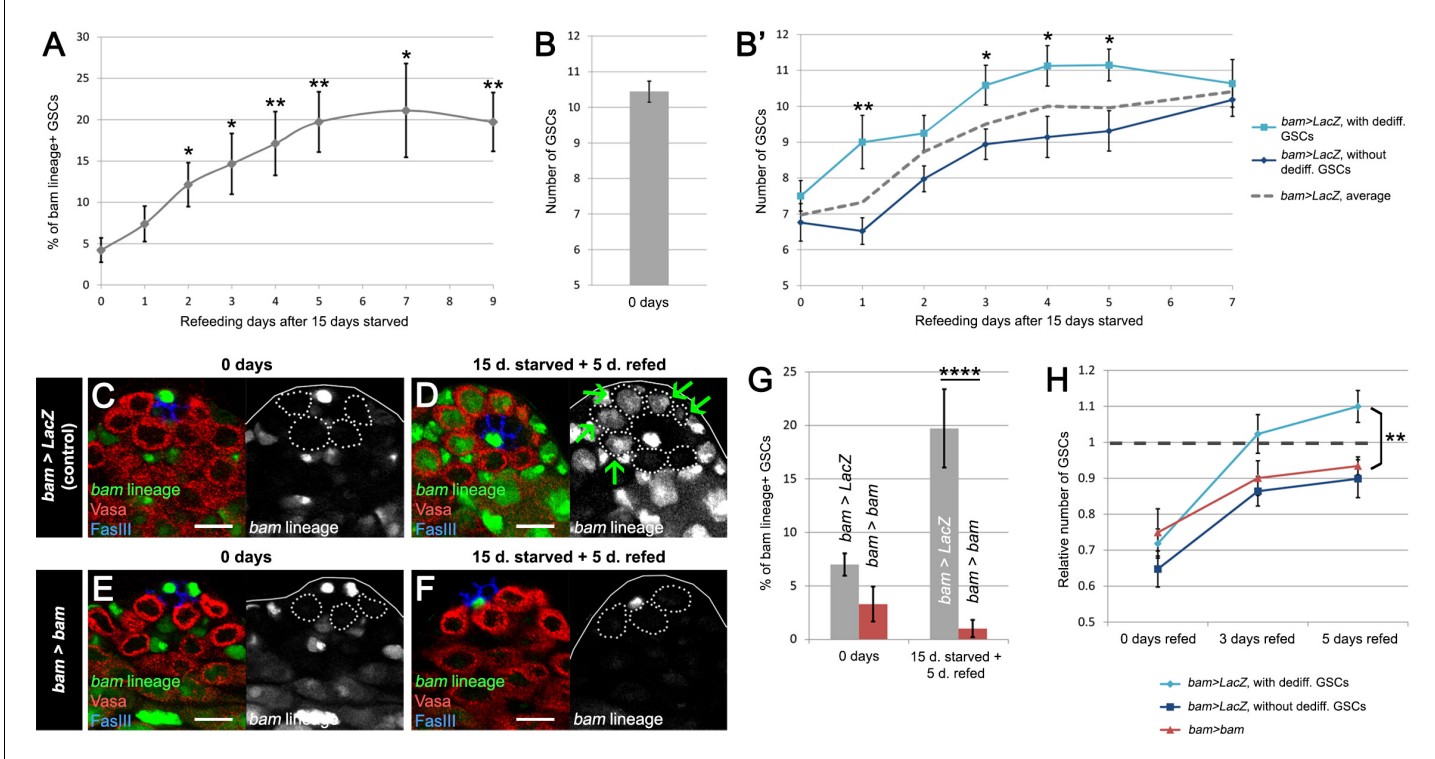

**Figure 2.** *bam*-lineage dedifferentiation accelerates the recovery of the GSC pool after starvation. (**A**) 9-day time course of the proportion of *bam*-lineage dedifferentiated GSCs during refeeding following 15 days of starvation in control *bam > LacZ* testes. (**B**) The total number of GSCs in *bam > LacZ* testes at 0 days prior to starvation and refeeding. (**B'**) Using the same data set as in (**A**), we counted the average total number of GSCs in all testes at each time point (dashed line). We then analyzed this data set according to whether *bam*-lineage dedifferentiated GSCs were present (light blue line) or absent (dark blue line). We observed two key features. First, there were significantly more total GSCs at nearly every time point in testes that contained *bam*-lineage dedifferentiated GSCs (light blue line). Second, the original 0 day total GSC number was recovered much faster in testes with *bam*-lineage GSCs, at 3 days of refeeding. By contrast, testes lacking *bam*-lineage GSCs required 7 days to fully recover the GSC pool (dark blue line). (**C–F**) Examples of testes from control *bam > LacZ* (**C,D**) and *bam > bam* (**E,F**) flies at 0 days and after 15 days of starvation and 5 days of refeeding. The *bam* lineage is labeled in green, the germline (Vasa) is red and the niche (FasIII) is blue. GSCs are outlined by the dashed line. There are no GSCs derived from *bam*-lineage germ cells in the control or *bam > bam* testis at 0 days (**C,E**). After 15 days starvation and 5 days refeeding, the control contains GSCs positive for the *bam*-lineage (**D**, arrows). However, at the same time point, there are no *bam*-lineage positive GSCs in the *bam > bam* testis (**F**). (**G**) Proportion of *bam*-lineage cells in the GSC pool in the aforementioned genotypes and time points. After starvation and refeeding, there is a significant reduction in the proportion of *bam*-lineage GSCs in *bam > bam* testes (red bar) compared to the control (gray bar). (**H**) Relative number of GSCs after 0, 3 or 5 days of refeeding following starvation. Light blue line indicates control *bam > LacZ* testes containing *bam*-lineage GSCs, whereas the dark blue line represents control testes lacking *bam*-lineage dedifferentiation. The red line represents *bam > bam* testes where dedifferentiation was blocked. Note that blocking dedifferentiation (red line) significantly delays the recovery of the GSC pool compared to testes with *bam*-lineage dedifferentiation (light blue line), in a manner similar to what happens in the control testes lacking dedifferentiated GSCs (dark blue line). Scale bars represent 10 μm. Bars on charts represent mean ±SE. *p<0.05, **p<0.01, ****p<0.0001.
DOI: https://doi.org/10.7554/eLife.36095.006

The following figure supplement is available for figure 2:

**Figure supplement 1.** Centrosome mis-orientation is not increased in *bam*-lineage GSCs compared to lineage-negative GSCs in aging and after challenging conditions.
DOI: https://doi.org/10.7554/eLife.36095.007

after 15 days of starvation) and a full recovery of the GSC pool after 5 days of refeeding (*Figure 2B, B'* and *Supplementary file 1*). In accordance with previous studies, we found that after the 15-day protein starvation period, the percentage of dedifferentiated GSCs did not increase compared to day 0 (compare *Figure 2A–G* and [*Yang and Yamashita, 2015*; *McLeod et al., 2010*]). However, after the 5 day refeeding period, this proportion significantly increased from 4.2% at day 0 after refeeding to 19.7% at day five after refeeding and did not increase further after longer refeeding times (*Figure 2A* and *Supplementary file 1*). We note that this increase in dedifferentiation during

the refeeding period strongly correlates with the recovery in the size of the GSC pool (*Figure 2B'*, gray dashed line).

Among the *bam > LacZ* controls, we observed variable numbers of *bam*-lineage dedifferentiated GSCs per testis. Most testes contained both GFP-negative wild type and GFP-positive dedifferentiated GSCs at the niche, however, some testes did not contain any dedifferentiated GSCs and some testes contained all dedifferentiated GSCs. We hypothesized that testes with more dedifferentiated GSCs had a faster recovery of the GSC pool after starvation and refeeding. Comparing testes with at least one dedifferentiated GSC (*Figure 2B'*, light blue line) to those with 0 dedifferentiated GSCs (*Figure 2B'*, dark blue line), we found that the former fully recovered the pool of GSCs after 3 days, while the latter required significantly longer, up to 7 days. Although we cannot exclude other variables such as germ cell death, GSC loss, and GSC gain through symmetric renewal, this correlation strongly suggests that *bam*-lineage dedifferentiation accelerates the recovery of the GSC pool under challenging conditions.

Thus, we hypothesized that preventing dedifferentiation using *bam > bam* flies would mimic this delay in recovery of the GSC pool observed in control flies lacking dedifferentiation. In *bam > bam* testes, the proportion of dedifferentiated GSCs did not increase (0% at day 0 after refeeding and 1.8% at day 5 after refeeding), in contrast to the *bam > LacZ* controls (*Figure 2C–G* and *Supplementary file 1*). Indeed, we observed that blocking dedifferentiation (i.e., *bam > bam*) retards the recovery of the GSC pool in a similar manner to control *bam > LacZ* testes devoid of *bam*-lineage dedifferentiation (*Figure 2H*, compare red to dark blue line). Compared to *bam > LacZ* testes with dedifferentiated GSCs, both *bam > bam* testes and *bam > LacZ* testes lacking dedifferentiated GSCs were significantly delayed in the recovery of the GSC pool (*Figure 2H*, compare red and dark blue lines to the light blue line). Centrosome mis-orientation in GSCs increases during aging and after irradiation (*Cheng et al., 2008*). We found that both dedifferentiated GSCs and their 'wild type' siblings displayed equally high rates (~35–40%) of centrosome mis-orientation after starvation and refeeding (*Figure 2—figure supplement 1*, third set of bars), similar to maximal rate of 40% reported in a previous study (*Cheng et al., 2008*). These results suggest that centrosome mis-orientation was not specific to *bam*-lineage dedifferentiated GSCs. Taken together, these results suggest that dedifferentiation of the *bam*-lineage promotes recovery of the GSC pool after challenging conditions.

## *bam*-lineage dedifferentiation preserves both the GSC pool and spermatogenesis during chronic challenging conditions

Our results suggest a functional role of dedifferentiation in maintaining a robust GSC pool during transient insults or challenging conditions. We further speculated that dedifferentiation could have additional biological roles when such conditions become chronic. We wondered whether males always housed with (and presumably mating with) females could force GSCs in the testis to cope with a higher demand for sperm production and thus accelerate the age-related reduction of the GSC pool. We note that while mating is a normal physiological event, it is stressful as mating significantly decreases both male and female lifespans (*Branco et al., 2017*; *Fowler and Partridge, 1989*). However, little is known about the effects of mating on the GSC pool in males. To determine this, we monitored the mitotic index in unmated vs continuously mated *bam > LacZ* males after 15 days of adulthood. We found significantly more GSCs in M-phase in testes from mated males compared to age-matched unmated males (*Figure 3—figure supplement 1A* and *Supplementary file 2*). We next assessed if both GFP-negative wild type and GFP-positive dedifferentiated GSCs in these two conditions had similar M-phase distribution. Indeed, in mated males there were more wild type and dedifferentiated GSCs in M-phase compared to unmated males (*Figure 3—figure supplement 1B* and *Supplementary file 2*). These data suggest that there is a systemic or non-autonomous effect of mating on GSC proliferation.

In a second set of experiments, we addressed whether mating increased dedifferentiation of the *bam* lineage. We aged *bam > LacZ* and *bam > bam* flies for 45 days in the constant presence of wild type *Oregon*[R] females. The rate of dedifferentiation in control *bam > LacZ* testes significantly increased from 7.0% at 0 days to 30.8% at 45 days when males are housed with females (*Figure 3— figure supplement 1C* and *Supplementary file 1*). As expected, the rate of dedifferentiation upon mating does not increase in *bam > bam* flies (*Figure 3—figure supplement 1C* and *Supplementary file 1*). Moreover, while the relative number of GSCs is not diminish in testes from

mated *bam > lacZ* controls, suggesting the pool is preserved, it is significantly (18%) smaller in mated *bam > bam* flies (*Figure 3—figure supplement 1D* and *Supplementary file 1*). Therefore, *bam*-lineage dedifferentiation is important in maintaining the size of the GSC pool under mating conditions. This result is in contrast to that obtained in unmated conditions (see *Figure 1G*).

Since both GSC proliferation and *bam*-lineage dedifferentiation are significantly increased upon mating, we speculated that this would affect GSC dynamics in the niche by increasing GSC turnover. Specifically we hypothesized that neutral GSC clones should be lost more frequently in testes from mated males compared to unmated ones. To test this, we generated neutral MARCM clones that expressed GFP but were otherwise wild type and measured clone residency in the niche at 2 days post clone induction (dpci) to establish clone induction frequency and at 15 dpci to monitor clone retention. As expected, clones were generated at equal frequency in both unmated and mated conditions (*Figure 3—figure supplement 1E*, first set of bars). However, at 15 dpci, there was a significant difference in clone retention with fewer clones retained in mated testes (*Figure 3—figure supplement 1E*, second set of bars). These results are consistent with the model in which increased proliferation of GSCs and outcompetition of resident GSCs by dedifferentiating germline cells alter stem cell dynamics in the testis niche.

After establishing the stressful effects of starvation and mating, we designed an aging protocol that might resemble sub-optimal conditions faced by flies in the wild, including intermittent access to food and to mating partners. This protocol consists of cycles of 6 days of protein starvation, which is sufficient to induce a significant reduction in the GSC pool (*Yang and Yamashita, 2015*), followed of 4 days of refeeding. In each cycle, during the last 2 days of the 4 days of refeeding, we added virgin females to the vials, which were subsequently removed for the next cycle. Our 10-day cycle of starvation and refeeding allowed us to accommodate 4 cycles of challenging conditions (see Materials and methods).

In control testes, after 4 cycles of starvation, refeeding and mating, we observed a significant increase in the rate of *bam*-lineage dedifferentiation, from 6.9% at day 0% to 43.9% at day 41 (*Figure 3A,B,E* and *Supplementary file 1*). As expected in *bam > bam* testes, the rate of dedifferentiation was completely blocked, 3.3% at day 0% to 2.1% at day 41 (*Figure 3C,D,E* and *Supplementary file 1*). Importantly, when we blocked dedifferentiation, the relative number of GSCs was significantly reduced by 33% compared to the controls after four cycles (*Figure 3F* and *Supplementary file 1*). These results indicate that, under these chronic challenging conditions, *bam*-lineage dedifferentiation is important to preserve the GSC pool.

We speculated that the reduced number of GSCs in testes lacking *bam-Gal4* lineage dedifferentiation might impact spermatogenesis. Indeed, after 2 and 4 cycles of starvation and refeeding, *bam > bam* testes were thinner and appeared to lack cells in intermediate stages of spermatogenesis (*Figure 4B,D,F*) when compared to matched *bam > lacZ* controls (*Figure 4A,C,E*). This phenotype is specific to chronic challenging conditions, because testes from 45 day old, fed and unmated flies displayed no apparent differences between the two genotypes (*Figure 4G,H*). We quantified the number of early-stage spermatogonia in the two genotypes: goniablasts (or 1-cell), 2-cell, 4-cell and 8-cell cysts. At 0 days, the number of gonia at each stage is indistinguishable between *bam > LacZ* and *bam > bam* flies (*Figure 4I* and *Supplementary file 3*), while as noted above, the absolute number of GSCs was distinct between the two genotypes. However, after 4 cycles of starvation, refeeding and mating, the number of gonia was significantly reduced up to 33% at each gonial stage when dedifferentiation was blocked (*Figure 4J* and *Supplementary file 3*). The reduced number of gonia in *bam > bam* testes is unlikely to be a direct result of the *bam* mis-expression, as the *bam-Gal4* line used is only active in the 4- and 8-cell stage, and *bam-Gal4* activity disappears once the cells have dedifferentiated (*Figure 1—figure supplement 2*). Instead, we observed a decrease in all the stages of spermatogenesis, including the pre-meiotic cysts, with a low-magnification inspection (*Figure 4A–F*).

## *bam*-lineage dedifferentiated GSCs are more proliferative than their siblings

The reduced number of gonia in *bam > bam* testes after four cycles (*Figure 4J*) could result from the reduced GSC pool in these testes (*Figure 3F*). However, an alternative explanation is that *bam > bam* testes lack the dedifferentiated GSCs found in control *bam > lacZ* testes and that these dedifferentiated GSCs have a higher proliferation rate than 'wild type' siblings and produced more

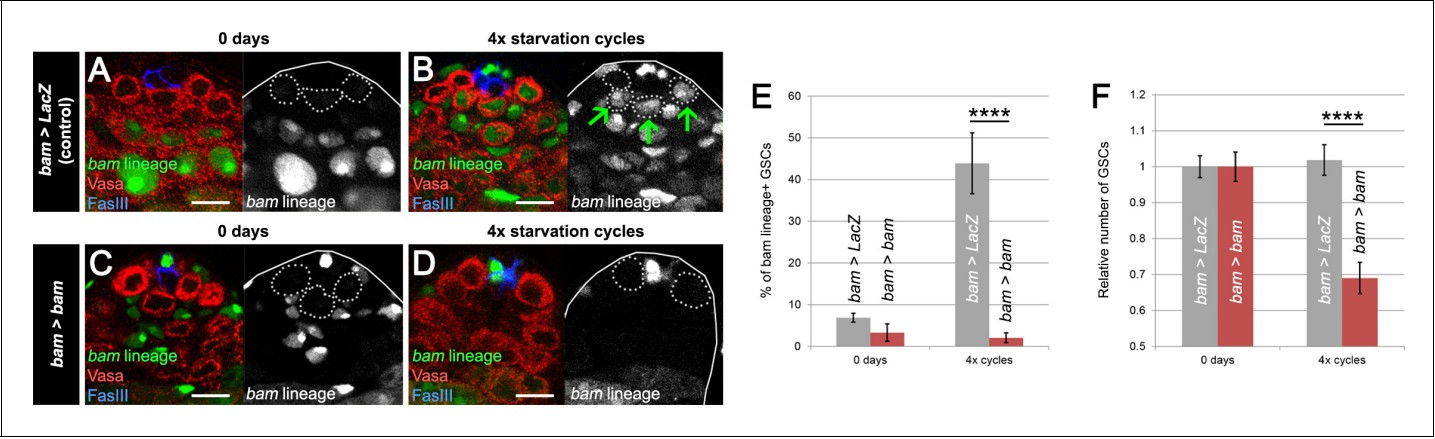

**Figure 3.** *bam*-lineage dedifferentiation is required for maintaining a robust GSC pool under chronic challenging conditions. (**A–D**) Representative images of the testis stem cell niche in control *bam > LacZ* (A,B) and *bam > bam* (C,D) testes at 0 days (A,C) and after 4 cycles of starvation, refeeding and mating (B,D). The *bam* lineage is labeled in green, the germline (Vasa) is red and the niche (FasIII) is blue. GSCs are outlined by the dashed line. After four cycles, *bam > LacZ* testes contain several *bam*-lineage GSCs (B, arrows). In the same conditions, *bam > bam* testes have fewer GSCs and none of them are dedifferentiated (D). (**E**) Proportion of *bam*-lineage GSCs in the indicated genotypes and conditions. The proportion of *bam*-lineage GSCs soars to nearly 50% in control testes (gray bars), and this process is completely abrogated in *bam > bam* testes (red bars). (**F**) The relative number of GSCs in control *bam > LacZ* and *bam > bam* testes. After 4 cycles of challenging conditions, there is a significant decline (33%) in the GSC pool in testes where dedifferentiation is prevented. Scale bars represent 10 μm. Bars represent mean ± SE. ****p<0.0001.
DOI: https://doi.org/10.7554/eLife.36095.008

The following figure supplement is available for figure 3:

**Figure supplement 1.** *bam*-lineage dedifferentiation is required for maintaining a robust GSC pool under conditions of continuous mating.
DOI: https://doi.org/10.7554/eLife.36095.009

gonial offspring. To assess whether *bam*-lineage dedifferentiated GSCs are more proliferative than their lineage-negative, wild type siblings, we directly compared the number of offspring of GFP-positive GSCs to those from GFP-negative GSCs in the same *bam > lacZ* control testis. We divided the number of spermatogonia at each stage by the number of labeled GSCs that were producing them. This experiment is analogous to a clonal analysis, but here we scored the contribution of an entire type of GSC rather than of a single GSC. Strikingly, we observed that after 4 cycles of starvation, dedifferentiated GSCs contribute up to 45% more offspring than their wild type siblings in the same testis (*Figure 5A* and *Supplementary file 3*). Additionally, we directly measured proliferation by scoring the proportion of GSCs positive for the S-phase marker EdU and the M-phase marker phospho-Histone3 (pH3). We found that there were significantly more *bam*-lineage positive GSCs in S-phase and in M-phase compared to *bam*-lineage negative GSCs (*Figure 5B,C*). Centrosome mis-orientation in GSCs slows the rate of proliferation during aging (*Cheng et al., 2008*). However, both *bam*-lineage-positive and lineage-negative GSCs had equally high rates of centrosome mis-orientation after 4 cycles of starvation and refeeding (*Figure 2—figure supplement 1*, fourth set of bars). These data suggest that in some challenging conditions, factors independent of centrosome orientation may regulate GSC proliferation rates.

### The JNK pathway is necessary for *bam*-lineage spermatogonial dedifferentiation

We reasoned that JNK signaling could be promoting spermatogonial dedifferentiation because of its established roles in cellular reprogramming and stress responses. An evolutionarily conserved kinase cascade, the JNK pathway plays essential roles in regeneration of numerous organs, including imaginal discs and intestine, and can trigger changes in cell identity and transdifferentiation (*Jiang et al., 2009*; *Bergantiños et al., 2010*; *Herrera et al., 2013*; *Smith-Bolton et al., 2009*; *Sun and Irvine, 2014*; *Herrera and Morata, 2014*; *Lee et al., 2005*; *Gettings et al., 2010*). JNK signaling is activated in a variety of stress responses; it is detected in the somatic support cells in the testis during protein starvation and contributes non-autonomously to the maintenance of the GSC pool (*Yang and Yamashita, 2015*).

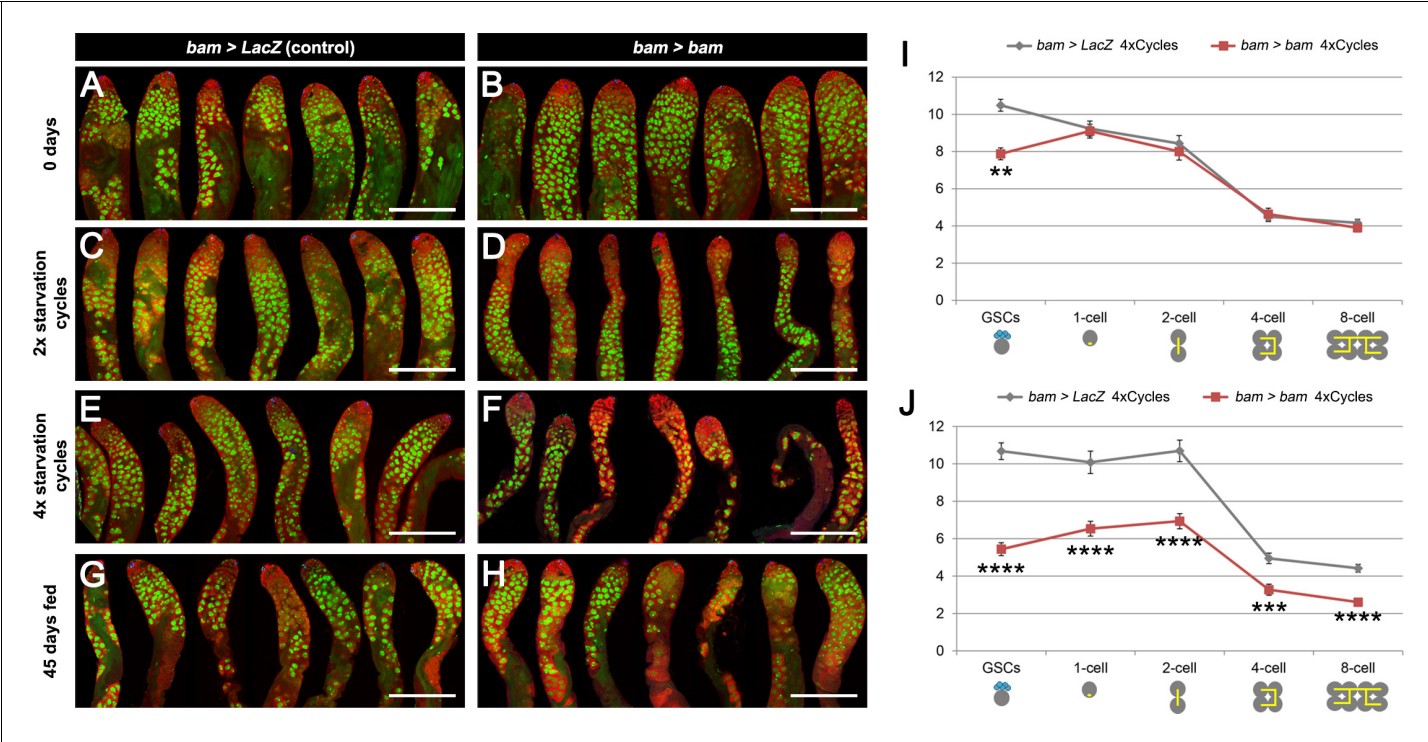

**Figure 4.** Spermatogenesis is compromised under chronic challenging conditions when *bam*-lineage dedifferentiation is inhibited. (A–H) Representative images of control *bam > LacZ* (A,C,E,G) and *bam > bam* (B,D,F,H) testes at 0 days (A,B), after 2 (C,D) or 4 (E,F) cycles of challenging conditions or in 45-day-old fed unmated males (G,H). The *bam* lineage is labeled in green and the germline (Vasa) is red. Note when preventing dedifferentiation and subjecting the animals to challenging conditions, the testes appear involuted, and visual inspection revealed fewer transit-amplifying spermatogonia and fewer pre-meiotic spermatocytes (D,F). These phenotypes were not observed in 45-day-old fed unmated *bam > bam* testes (H), indicating that they do not arise simply as a result of aging. (I–J) Quantification of the number of GSCs and 1-, 2-, 4- and 8-cell spermatogonial cysts at 0 days (I) and after 4 cycles of challenging conditions (J) in *bam > LacZ* (gray line) and *bam > bam* (red line) males. Spermatogonial staging was determined using an antibody to αSpectrin, which marks the fusome/spectrosome. After four cycles, there were significantly fewer spermatogonia of each class in testes where dedifferentiation was inhibited (*bam > bam*). Scale bars represent 200 µm. Data points represent mean ±SE. **p<0.01, ***p<0.001, ****p<0.0001.
DOI: https://doi.org/10.7554/eLife.36095.010

We were unable to detect real-time JNK activity reporters *puckered* (*puc*)-*LacZ* (*Martín-Blanco et al., 1998*) or *TRE-GFP* (*Chatterjee and Bohmann, 2012*) in the germline during or after starvation (data not shown). Since the frequency of dedifferentiation is on average 2.2 GSCs per testis during 5 days of refeeding (*Supplementary file 1*), the chances for detecting this JNK activation are low, especially if it is transient and/or if only low levels are required. For this reason, we decided to use a more sensitive assay: lineage labeling of *puc-Gal4*-positive cells. *puc* is a transcriptional target and repressor of the pathway (*Martín-Blanco et al., 1998*). Similar to the *bam-Gal4* lineage labeling, Flp under the control of *puc-Gal4* recombines an *ubiP63E-FRT-stop-FRT-GFP* cassette so that cells that have expressed *puc* in the past, even at low levels, will become permanently labeled by GFP (genotype: *puc > GFP*). Two *puc-Gal4* lines with independent origins were used for these experiments (see Materials and methods), showing similar results.

At 0 days old, testes from *puc > GFP* males had GFP expression in a fraction of hub cells. However, GFP expression was largely absent from somatic cyst cells and the germline, with 20% and 12% of testes, respectively, labeled at 0 days in fed conditions (*Figure 6A,F,F'*). Testes from *puc > GFP* males that were maintained under fed conditions and aged for 15 or 20 days showed a similar trend of low rates of labeling (*Figure 6B,C,F,F'*). After 15 days of protein starvation, 51% of *puc > GFP* testes displayed GFP labeling of somatic cyst cells, while GFP expression in the germline was observed in less than 11% of the cases (*Figure 6D,F,F'*). This increase of JNK signaling in the somatic cells after starvation is consistent with a previous study (*Yang and Yamashita, 2015*). After

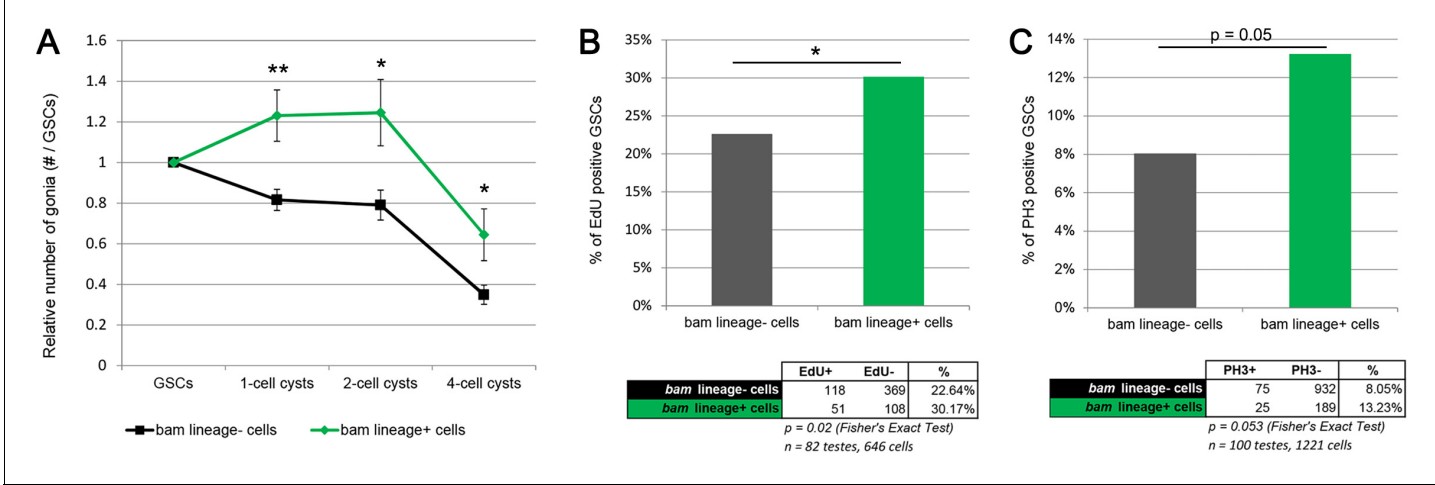

**Figure 5.** *bam*-lineage GSCs are more proliferative than their lineage-negative siblings. (**A**) Relative contribution to 1-, 2- or 4-cell spermatogonial cysts of *bam*-lineage GSCs (green line) and lineage-negative sibling GSCs (black line) in the same *bam > LacZ* testis (see Materials and methods for details). *bam*-lineage GSCs have significantly more spermatogonial offspring than lineage-negative sibling GSCs. (**B**) EdU incorporation in *bam*-lineage GSCs (green bar) and lineage-negative sibling GSCs (gray bar) in the same *bam > LacZ* testis. There are significantly more EdU-positive lineage-positive GSCs than EdU-positive lineage-negative sibling GSCs. (**C**) pH3 labeling in *bam*-lineage GSCs (green bar) and lineage-negative sibling GSCs (gray bar) in the same *bam > LacZ* testis. There are significantly more pH3-positive *bam*-lineage GSCs than pH3-positive lineage-negative sibling GSCs. The raw data are presented in the chart below the graph. Data points represent mean ±SE. *p<0.05, **p<0.01.
DOI: https://doi.org/10.7554/eLife.36095.011

15 days of starvation and 5 of refeeding, 82% of *puc > GFP* testes had GFP labeling of somatic cells, and 55% now displayed germline labeling as well, including GSCs (*Figure 6E–F'*). These results indicate that germline cells acquire JNK activity specifically during the refeeding phase, when the burst of dedifferentiation takes place (*Figure 2A*).

We analyzed germline labeling at 2 and 3 days post refeeding to gain insight into how spermatogonial cysts dedifferentiate (*Figure 6F*). Because prior work has documented fragmenting of 4- and 8-cell gonia prior to dedifferentiation into GSCs (*Brawley and Matunis, 2004*; *Sheng et al., 2009*; *Cheng et al., 2008*; *Sheng and Matunis, 2011*), we predicted that we would observe a majority of testes with only gonia but not GSCs labeled with *puc > GFP* at these time points. Of the 97 testes scored at 2 and 3 days post refeeding, 17 (17.5%) had germline labeling (defined as any GSC, 1-, 2-, 4-, 8- and/or 16-cell gonia expressing *puc > GFP*). Of these 17 testes, most (n = 15) had at least 1 GSC labeled as well as various gonia; only two testes had gonia but no GSCs labeled. We speculate that high number of cases where an entire germline lineage is labeled results from the considerable time (>24 hr) required to activate *puc*, recombine the lineage-tracing cassette and induce GFP expression. Our observations are consistent with JNK-activated gonia breaking apart and liberating gonial cells to dedifferentiate into GSCs. However, we cannot rule out the possibility that during refeeding JNK signaling is directly activated in resident GSCs that have not dedifferentiated. In support of our model that spermatogonia fragment and revert to the GSC state after autonomous JNK activation, we observed one testis at 3 days of refeeding where a 4-cell gonia labeled with *puc > GFP* appeared to fragment into a 3-cell gonia and a GSC at the niche (*Figure 6—figure supplement 1*). Each germ cell in the 3-cell gonia had a dot spectrosome, characteristic of fragmenting germ cells during dedifferentiation and very early germ cells (GSCs and gonialblasts) in wild type (*Cheng et al., 2008*; *de Cuevas et al., 1997*).

To functionally test the relevance of JNK activity in spermatogonial dedifferentiation, we blocked its activity in the *bam-Gal4* lineage by mis-expressing the JNK inhibitor *puc* or a dominant negative form of the JNK *basket (bsk)*. Concomitantly, we lineage traced *bam-Gal4* cells. If JNK activity is necessary for dedifferentiation, we predict that blocking its activity would reduce dedifferentiation upon refeeding. Indeed, after one cycle of starvation and refeeding, the increase in *bam*-lineage GSCs was no longer detectable in *bam > puc* or *bam > bsk^DN* testes compared with *bam > LacZ* controls (*Figure 6G–K*). We note that neither mis-expression of *puc* nor *bsk^DN* appears to adversely affect

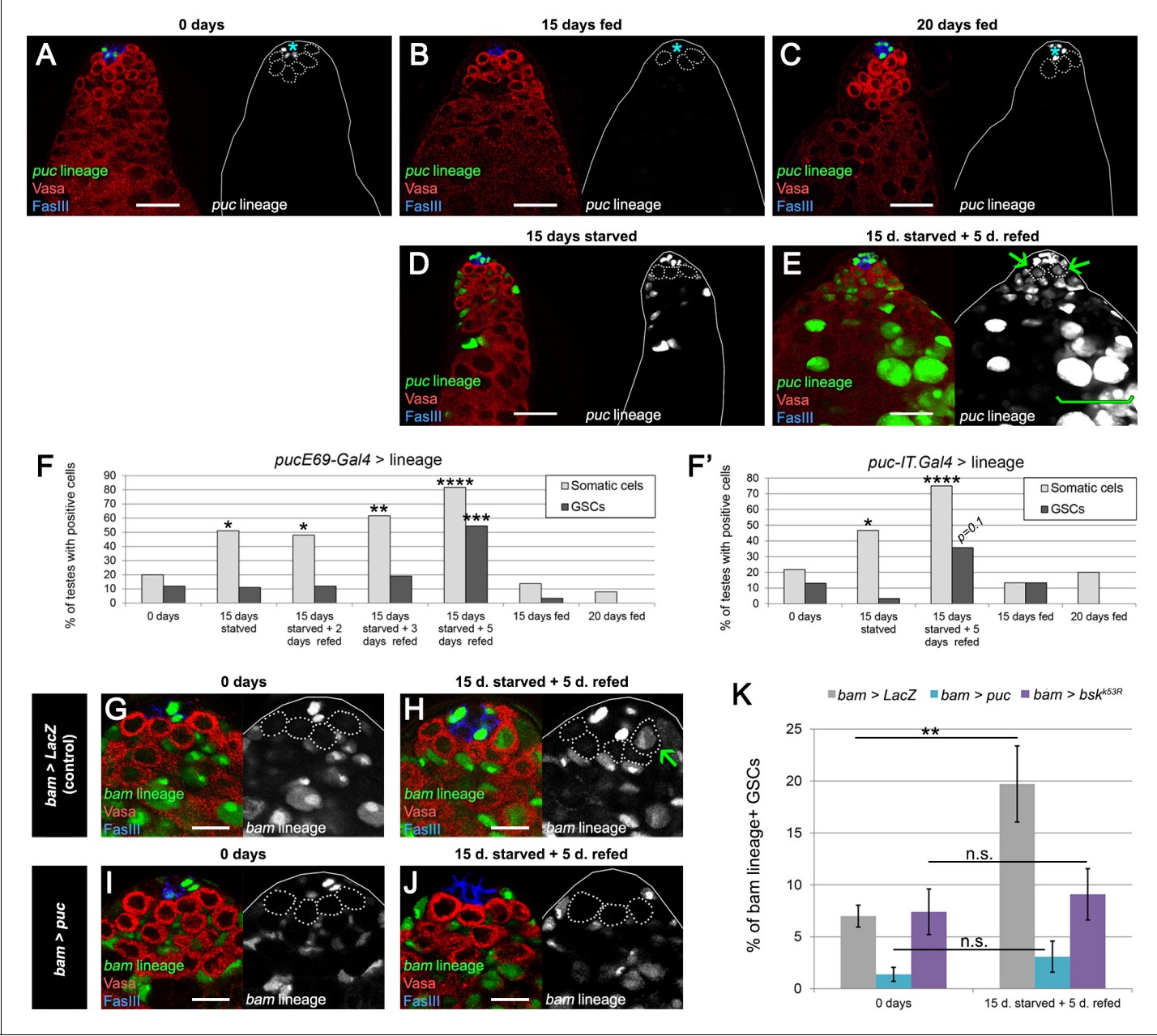

**Figure 6.** JNK pathway activity is required for *bam*-lineage spermatogonial dedifferentiation. (A–E) Representative images of testes where *puckered*[E69] *(puc)-Gal4* was lineage-traced at 0 days fed (A), 15 days fed (B), 20 days fed (C), after 15 days starved (D) and after 15 days starved and 5 days refed (E). Under fed conditions, *puc*-lineage cells were restricted to some niche cells (asterisk) (A–C). After 15 days of starvation, most testes had *puc*-lineage labeling of somatic cells (quantified in F) (D). Only during the refeeding phase were *puc*-lineage germline cells observed (quantified in F) (E). Arrows in E mark *puc*-lineage-positive GSCs and the bracket in E marks *puc*-lineage-positive spermatogonia. (F, F') Graphs indicating the percentage of testes that had *puc*[E69]*-Gal4-* (F) or *puc-IT.Gal4-* (F') lineage positive somatic cells (light gray bars) or germline cells including GSCs (dark gray bars) at the time points indicated in A-E, as well as at 2 and 3 days of refeeding in (F). (G–J) Representative images of the testis stem cell niche in control *bam-Gal4, UAS-LacZ (bam > LacZ,* G, H) or *bam-Gal4, UAS-puc (bam > puc,* I, J), where we blocked JNK signaling, at 0 days (G,I) of after 15 days of starvation and 5 days of refeeding (H,J). The *bam* lineage is labeled in green, the germline (Vasa) is red and the niche (FasIII) is blue. GSCs are outlined by the dashed line. There are no GSCs derived from the *bam*-lineage cells in the control or *bam > puc* testes at 0 days (G, I). After starvation and refeeding, the control (*bam > LacZ*) contains GSCs positive for the *bam*-lineage (indicating that they arose through dedifferentiation) (H, arrow). However, under the same conditions, there are no *bam*-lineage positive GSCs in the *bam > puc* testis (J), indicating that JNK signaling is required for dedifferentiation. (K) Quantification of the rate of *bam*-lineage dedifferentiation among GSCs in controls (gray bars) and after repressing the JNK pathway with either *UAS-puc* (blue bars) or a dominant negative form of *basket (UAS-bsk*[K53R]*)* (purple bars). Both transgenes are able to block dedifferentiation, mimicking the

*Figure 6 continued on next page*

*Figure 6 continued*

effects of *bam* mis-expression. Scale bars in A-E represent 20 µm, while in G-J they represent 10 µm. Bars in the graphs represent mean ±SE. *p<0.05, **p<0.01, ***p<0.001, ****p<0.0001.

DOI: https://doi.org/10.7554/eLife.36095.012

The following figure supplements are available for figure 6:

**Figure supplement 1.** Example of fragmentation of 4-cell spermatogonia that experienced JNK activation.

DOI: https://doi.org/10.7554/eLife.36095.013

**Figure supplement 2.** *puc* or *bsk*<sup>DN</sup> mis-expression does not perturb germ cell differentiation.

DOI: https://doi.org/10.7554/eLife.36095.014

germ cell differentiation as robust spermatogonia and spermatocytes expressing *puc* or *bsk*<sup>DN</sup> survive and continue to progress towards meiotic stages (*Figure 6—figure supplement 2*). Taken together, these results indicate that JNK pathway activity is autonomously necessary for germline cells to undergo dedifferentiation.

## Discussion

While spermatogonial dedifferentiation increases during aging (this study and (*Cheng et al., 2008*)), the biological role of this process has remained unknown. Surprisingly, through a combination of genetic methodologies, we demonstrate that dedifferentiation of the *bam* lineage plays no role in maintaining the GSC pool during aging under standard conditions (abundant food and no females). Instead, we find that under normal but stressful conditions such as mating or under challenging conditions such as starvation and refeeding, dedifferentiation is important for both the quick recovery of the GSC pool in the short term and the preservation of the stem cell number in the long term. These results lead us to propose that *bam*-lineage dedifferentiation is akin to a regenerative response aimed to preserve the number of gonadal stem cells under adverse situations but is dispensable under optimal life conditions. This model is consistent with previous results demonstrating an increase in dedifferentiation after damage induced by irradiation (*Tetteh et al., 2016*; *van Es et al., 2012*; *Cheng et al., 2008*). Furthermore, our results suggest that spermatogonial cysts can fragment and the liberated germ cells migrate back to the niche to become functional GSCs, similar what was observed upon regeneration of the germline after GSC depletion (*Figure 7* and [*Brawley and Matunis, 2004*; *Sheng et al., 2009*]). Since germ cells from fragmented cysts have to compete against resident GSCs as well as resident somatic stem cells in order to re-occupy the niche, germ cells that successfully dedifferentiate must have increased competitive properties, which are currently not understood. Future work using live imaging will be needed to uncover the dynamics of this process.

Our results reveal a critical role of the JNK pathway in spermatogonial dedifferentiation. We have detected its activation during the refeeding phase after starvation and have proven its requirement in promoting this phenomenon (*Figure 7*). As mentioned above, we show that dedifferentiation is critical to maintaining a robust GSC pool during challenging conditions and as such is similar to a regenerative response. Recently, several studies have demonstrated the importance of the JNK pathway for proliferation, for triggering other signaling pathways, and for cellular reprogramming during regenerative events in imaginal discs (*Bergantiños et al., 2010*; *Herrera et al., 2013*; *Smith-Bolton et al., 2009*; *Sun and Irvine, 2014*; *Herrera and Morata, 2014*; *Lee et al., 2005*). It is unclear what JNK signaling regulates during spermatogonial dedifferentiation, but this phenomenon involves some of the features of regeneration, particularly reprogramming which reverts the cell identity to 'stemness'. Taken together with our results, these studies suggest that JNK activity may be a universal feature of regenerative responses in *Drosophila* whether it is to reconstruct an appendage or recover a pool of stem cells.

Similar to our results, two previous studies did not detect *bam*-lineage-traced, dedifferentiated GSC in the testis during protein starvation (*Yang and Yamashita, 2015*; *McLeod et al., 2010*). Therefore, there is consensus in the field that germline dedifferentiation does not occur during starvation. Our work shows that dedifferentiation occurs during the refeeding period with the maximal percentage of dedifferentiated GSCs being observed only after 5 days. We note that another group did not observe dedifferentiation during the refeeding period. However, they only scored this event

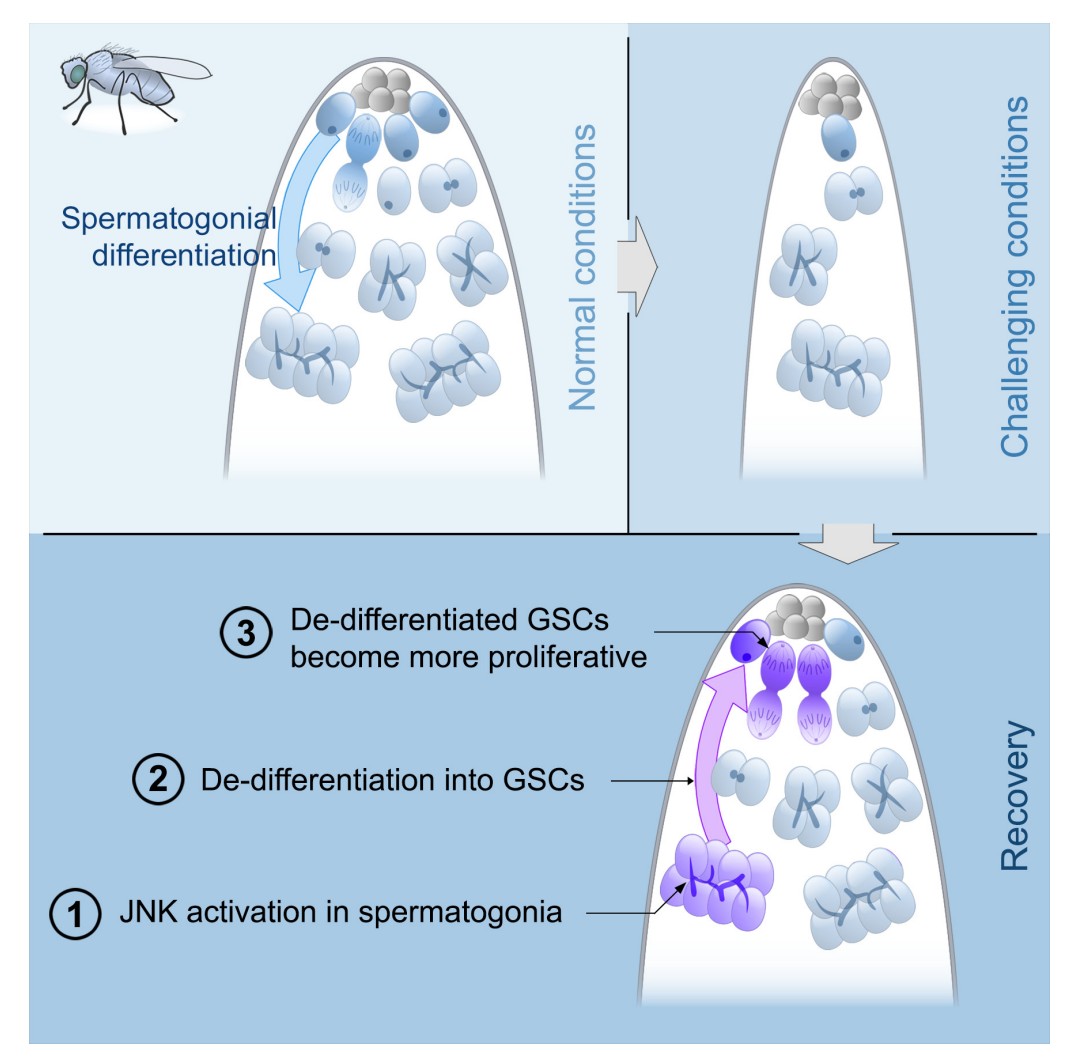

**Figure 7.** Model of spermatogonial dedifferentiation under chronic challenging conditions. Top left, under normal conditions, GSCs divide and their offspring undergo normal differentiation into spermatogonia. Top right, under challenging conditions like starvation, there is a significant reduction in the number of GSCs and their offspring. Bottom, during the recovery phase, JNK signaling is triggered in spermatogonia (number 1), these cysts break apart and the germ cells dedifferentiate into GSCs (number 2). Dedifferentiated GSCs have higher proliferation rates than their wild type sibling GSCs and produce more offspring (number 3).

DOI: https://doi.org/10.7554/eLife.36095.015

at 2 days after refeeding and not at later time points (*McLeod et al., 2010*). We think the discrepancy between our studies is due to the different lengths of the refeeding period.

One caveat of our results is the possibility that spurious activity of *bam-Gal4* in GSCs could be forcing the recombination of the GFP flip-out cassette and thus marking them indistinguishably from truly dedifferentiated cells. There are, however, three arguments against this being responsible for the increase in *bam*-lineage labeled GSCs: (1) we failed to detect real time expression of *bam-Gal4* in GSCs (*Figure 1—figure supplement 2*); (2) preventing dedifferentiation (*bam > bam*) blocked the increase in labeled GSCs in every scenario we tested; and (3) under challenging conditions, the GFP-positive GSCs were demonstratively more proliferative than their unlabeled siblings, a result that is inconsistent with spurious labeling.

Another caveat of our experiments is that the extent of dedifferentiation may be underscored because our labeling methodology precludes us from lineage-tracing very early germ cells; the driver we used (*bam-Gal4*) is restricted to 4- and 8-cell spermatogonia and does not label gonialblasts (i.e.,

1-cell) and 2-cell spermatogonia. To the best of our knowledge, there are no Gal4 lines specific to the lattermost cell types and at the same time excluded from GSCs. This limitation makes it impossible for us to track dedifferentiation of gonialblasts and 2-cell spermatogonia and likely causes undersampling of all dedifferentiation events. We note that examples of dedifferentiation from 2-cell gonia have been documented by live imaging (*Sheng and Matunis, 2011*); this study also showed that 'symmetric renewal' occurs when the gonialblast of a GSC-gonialblast pair swivels into the niche resulting in two stem cell offspring instead of 1 stem cell and 1 differentiating offspring. 'Reversion' occurs when spermatogonial cysts break apart and germ cells return to the niche to become functional GSCs (*Sheng and Matunis, 2011*). Both symmetric renewal and reversion could be at play in challenging conditions and may underscore why, following one cycle of starvation, the GSC pool does indeed recover albeit in a delayed fashion even when dedifferentiation is blocked (*Figure 2B'*).

The inability to track dedifferentiation of gonialblasts and 2-cell gonia, combined with the less than 100% efficiency of Flp/FRT, may underlie the increased mis-oriented centrosomes in *bam*-lineage negative GSCs under a variety of conditions tested here. Although we observed a significantly higher proportion of *bam*-lineage GSCs with mis-oriented centrosomes compared to lineage-negative GSCs at day 0, there was no statistically significant difference between these two populations after 45 days of 'normal' aging (*Figure 2—figure supplement 1*), consistent with a prior report (*Cheng et al., 2008*). Additionally, we assessed mis-oriented centrosomes after challenging conditions like 1 cycle of starvation and refeeding and 4 cycles of challenging conditions and found no difference in *bam*-lineage positive vs negative GSCs (*Figure 2—figure supplement 1*). It is possible that the lineage-negative GSCs are in fact dedifferentiated from gonialblasts and 2 cell gonia. While we cannot experimentally test this hypothesis, if both types of GSCs were in fact derived from dedifferentiated gonia, this could account for the roughly equal rates of mis-oriented centrosomes in both pools (*Figure 2—figure supplement 1*).

Despite similarly high rates of centrosome mis-orientation in *bam*-lineage dedifferentiated GSCs and lineage-negative siblings, the former proliferate faster. This result was unexpected, as it has been reported that dedifferentiated GSCs have decreased proliferation rates due to increased centrosome mis-orientation (*Cheng et al., 2008*). We speculate that downstream effectors of JNK signaling could be responsible for this increased cycling. The transient activity of JNK in germline cells that dedifferentiate could elicit a change in the epigenetic landscape by modulating *Polycomb*-group (*Pc*-g) and *trithorax*-group (*trx*-g) genes, as has been shown in several regenerative contexts (*Herrera and Morata, 2014*; *Lee et al., 2005*; *Roumengous et al., 2017*). Such epigenetic changes, possibly downstream of JNK/*Pc*-g or JNK/*trx*-g, may endow dedifferentiated GSCs with increased proliferation compared to their wild type siblings. For example, potential cell reprogramming of dedifferentiated GSCs may 'refresh' a stem cell's genetic landscape, compared with its wild type siblings that may have acquired genetic damage or imprinting. Future experiments will be necessary to directly test these hypotheses.

## Materials and methods

**Key resources table**

| Reagent type (species) or resource | Designation | Source or reference | Identifiers | Additional information |
|---|---|---|---|---|
| Strain, strain background (*Drosophila melanogaster*) | *Oregon-R (Ore)$^R$* | Bloomington Drosophila Stock Center (BDSC) | stock number: 5 | |
| Strain, strain background (*Drosophila melanogaster*) | *Ubi-p63E(FRT.STOP)Stinger* | Bloomington Drosophila Stock Center (BDSC) | stock number: 28282 | |
| Strain, strain background (*Drosophila melanogaster*) | *UAS-RedStinger* | Bloomington Drosophila Stock Center (BDSC) | stock number: 28281 | |
| Strain, strain background (*Drosophila melanogaster*) | *UAS-LacZ* | Bloomington Drosophila Stock Center (BDSC) | stock number: 3955 | |

*Continued on next page*

Continued

| Reagent type (species) or resource | Designation | Source or reference | Identifiers | Additional information |
|---|---|---|---|---|
| Strain, strain background (*Drosophila melanogaster*) | UAS-bsk$^{K53R}$ | Bloomington Drosophila Stock Center (BDSC) | stock number: 9311 | |
| Strain, strain background (*Drosophila melanogaster*) | puc$^{E69}$-Gal4 | Bloomington Drosophila Stock Center (BDSC) | stock number: 6762 | |
| Strain, strain background (*Drosophila melanogaster*) | puc-IT.Gal4 | Bloomington Drosophila Stock Center (BDSC) | stock number: 63509 | |
| Strain, strain background (*Drosophila melanogaster*) | UAS-puc | pmid: 9472024 | | |
| Strain, strain background (*Drosophila melanogaster*) | bam-Gal4:VP16 | pmid: 12571107 | | |
| Strain, strain background (*Drosophila melanogaster*) | UAS-bam:GFP | pmid: 12571107 | | |
| Strain, strain background (*Drosophila melanogaster*) | FRT$^{40A}$ | pmid: 26807580 | | |
| Strain, strain background (*Drosophila melanogaster*) | ywhsFlp$^{112}$; tubGal80,FRT$^{40A}$ | pmid: 26807580 | | |
| Antibody | goat anti-Vasa | Santa Cruz | catalog number: dC-13 | '1:200' |
| Antibody | mouse anti-Fas3 | DHSB | catalog number: 3A9 | '1:50' |
| Antibody | mouse anti-α-Spectrin | DSHB | catalog number: 7G10 | '1:50' |
| Antibody | mouse anti-γTubulin | Sigma | catalog number: T6557 | '1:100' |
| Antibody | 5-ethynyl-2'-deoxyuridine (EdU) | Invitrogen | catalog number: C10340 | '1:50' |
| Antibody | rabbit anti-phospho Histone H3-Ser10 | Millipore | catalog number: #06–570 | '1:200' |
| Software, algorithm | Fiji-ImageJ | pmid: 22743772 | | |
| Software, algorithm | Photoshop | Adobe | | |
| Software, algorithm | Excel | Microsoft | | |
| Software, algorithm | GraphPad Prism | GraphPad Software | | |
| Software, algorithm | Paint3D | Microsoft | | |

## *Drosophila* stocks

The following fly stocks were obtained from the Bloomington *Drosophila* Stock Center (BDSC): *Oregon-R (Ore)$^R$*, *Ubi-p63E(FRT.STOP)Stinger*, *UAS-RedStinger*, *UAS-LacZ*, *UAS-bsk$^{K53R}$*, *puc$^{E69}$-Gal4*, *puc-IT.Gal4*. Additionally, we used the following fly stocks: *UAS-puc* (**Martín-Blanco et al., 1998**); *bam-Gal4:VP16* (**Chen and McKearin, 2003**); *UAS-bam:GFP* (**Chen and McKearin, 2003**); *FRT$^{40A}$* and *y, w, hs-Flp$^{112}$; tub-Gal80, FRT$^{40A}$* (**Amoyel et al., 2016**).

## Antibodies

The primary antibodies used were: goat anti-Vasa (Santa Cruz, 1:200), mouse anti-FasIII (Developmental Studies Hybridoma Bank (DSHB), 1:50), mouse anti-α-Spectrin (DSHB, 1:50), mouse anti-γ Tubulin (Sigma, 1:100), rabbit anti-phospho-Histone3-Ser10 (Millipore, 1:200). 5-ethynyl-2′-deoxyuridine (EdU, Invitrogen) labeling was carried out as previously described (*Amoyel et al., 2014*).

## Aging conditions

In all the experiments, flies were raised at 25°C. For standard aging conditions, virgin males were collected and kept isolated from females. Flies were kept in vials with food at a density of 20 males per vial (1-inch width) and transferred into fresh new vials every 2 days.

Flies were maintained on standard fly food. For starvation (protein deprivation) periods, males were transferred into vials with 10% sucrose/1% agar, replaced by fresh ones every 2 days.

For aging in the presence of females (mated males), a maximum of 20 males were placed in the same vial with 40 young (no older than 1 week) wild type (*Oregon-R*) virgin females. Flies were transferred to new vials every 2 days, and the females were replaced by new young virgins every 2 weeks.

For aging in challenging conditions, we subjected flies to a regime of cycles of the following composition: 6 days of protein deprivation, followed by 2 days of refeeding in standard food and two additional days of refeeding in standard food in the presence of virgin wild type *Oregon-R* females in a 2:1 ratio of females to males. At the end of each cycle, females were discarded. For the last cycle, we extended the refeeding time one extra day, so the refeeding phase encompassed 5 days instead of 4, in order to collect testes with a degree of recovery comparable to the single-cycle experiments.

## Data analysis and statistics

We scored a cell as a GSC if it met these conditions: (1) it is Vasa-positive (Vasa is expressed only in germline cells in the testis); (2) it is a single cell that is not part of a spermatogonial cyst; (3) it makes direct contact with the niche (FasIII-positive cells). In some experiments, the second criterion was evaluated by assessing the presence of dot fusomes when stained with α-Spectrin, as this is a hallmark of GSCs.

In the course of our experiments, we realized that distinct genotypes have a different number of GSCs at 0 days (see *Figure 4I* for example). To compare total GSCs in each genotype over time, we normalized the total number of GSCs at each time point to that of the start (i.e., day 0). For this reason, most of the data are shown as 'relative GSC number', thus enabling direct comparisons between genotypes after the same treatment.

The percentage of dedifferentiation is calculated as (1) the proportion of GFP-positive (i.e., *bam*-lineage-positive) GSCs divided by the total number of GSCs in each individual testis and (2) each testis in a particular genotype was averaged.

The number of cysts at each stage in *Figures 4I, J* and *5A* was scored using the α-Spectrin antibody, which labels the fusome that connects all the cells in a spermatogonium.

The percentage of GSCs with mis-oriented centrosomes was calculated by means of an established methodology (*Cheng et al., 2008*).

Images were acquired on a Zeiss LSM 510 confocal microscope. Image analysis and quantifications were performed with Fiji-ImageJ (*Schindelin et al., 2012*) and Adobe Photoshop software. Fisher's exact tests were used for *Figures 5B, C* and *6F–F'* and *Figure 3—figure supplement 1A,B, E*. The rest of the statistical tests were performed with Student's t tests. Data were analyzed with Microsoft Excel and GraphPad Prism. A summary of results, including averages and sample sizes is included in *Supplementary file 1–3*. Paint3D was used in *Figure 6—figure supplement 1* to illustrate the position of the confocal slice in the z-stack as well as to indicate the position of the fragmenting germ cyst.

## Acknowledgements

We thank Yukiko Yamashita, Ruth Lehmann, Gines Morata, the Developmental Studies Hybridoma Bank and the Bloomington Stock Center for stocks and antibodies, and Hyung Don Ryoo for critical

reading of the manuscript. This work was supported by EMBO and HFSP LT000529-2015 fellowships (to SH), and NYSTEM N11G-292 and NIH R01 GM085075 (to EAB).

## Additional information

### Funding

| Funder | Grant reference number | Author |
|---|---|---|
| National Institute of General Medical Sciences | R01 GM085075 | Erika A Bach |
| European Molecular Biology Organization | | Salvador C Herrera |
| Human Frontier Science Program | LT000529-2015 | Salvador C Herrera |
| New York State Department of Health | NYSTEM N11G-292 | Erika A Bach |

The funders had no role in study design, data collection and interpretation, or the decision to submit the work for publication.

### Author contributions

Salvador C Herrera, Conceptualization, Resources, Data curation, Formal analysis, Funding acquisition, Validation, Investigation, Visualization, Methodology, Writing—original draft, Project administration, Writing—review and editing; Erika A Bach, Conceptualization, Resources, Supervision, Funding acquisition, Visualization, Methodology, Writing—original draft, Project administration, Writing—review and editing

### Author ORCIDs

Erika A Bach (iD) http://orcid.org/0000-0002-5997-4489

### Decision letter and Author response

Decision letter https://doi.org/10.7554/eLife.36095.021
Author response https://doi.org/10.7554/eLife.36095.022

## Additional files

### Supplementary files

• Supplementary file 1. Raw data collected for this work. Data are presented as the mean ± S.E.
DOI: https://doi.org/10.7554/eLife.36095.016

• Supplementary file 2. Raw data of proliferation markers analyzed for this work.
DOI: https://doi.org/10.7554/eLife.36095.017

• Supplementary file 3. Raw data collected for this work. Data are presented as mean ± S.E.n.a is not applicable
DOI: https://doi.org/10.7554/eLife.36095.018

• Transparent reporting form
DOI: https://doi.org/10.7554/eLife.36095.019

### Data availability

All data generated or analyzed during this study are included in the manuscript and supporting files. Source data files have been provided for supplementary files 1–3.

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
