## [Decision Letter]

Thank you for submitting your article "*JNK* triggers dedifferentiation during chronic stress to maintain the germline stem cell pool in the *Drosophila* testis" for consideration by *eLife*. Your article has been reviewed by three peer reviewers, including Yukiko M Yamashita as the Reviewing Editor and Reviewer #1, and the evaluation has been overseen by Marianne Bronner as the Senior Editor. The following individual involved in review of your submission has agreed to reveal their identity: Michael Buszczak (Reviewer #3).

The reviewers have discussed the reviews with one another and the Reviewing Editor has drafted this decision to help you prepare a revised submission.

As is provided below in reviewers' individual comments, all felt that this study provides significant insights into biological significance of dedifferentiation, warranting publication in *eLife*. However, there are several issues that need to be addressed prior to acceptance. Since reviewers' individual comments are not conflicting and in agreement, we decided to provide full review comments as is such that the authors can see the full spectrum of the basis of our discussion.

The outline of essential revisions we ask is the following:

1) There are strong concluding statements throughout the manuscript that are not necessarily backed up by the experimental evidence. (Also, there are several cases where the authors make rather strong statements in the Results, only to back off of them in the Discussion. These should be removed.)

- There are several concerns regarding Bam-gal4-mediated lineage tracing of dedifferentiated GSCs. Although Bam-gal4-mediated lineage tracing indeed marks dedifferentiation, it does not mark *all* of dedifferentiated cells, because 2-cell SGs do not express *bam-gal4*. However, 2-cell SGs are considered to dedifferentiate more than 4- and 8-cell SGs due to their proximity to the niche. Thus, one would have to assume that there are many more dedifferentiated GSCs that are negative for lineage-marker than those positive for lineage-marker. Throughout the Results section, the authors do not sufficiently acknowledge this (highly likely) possibility, and draw strong conclusions, essentially assuming that all of linage-negative cells have never undergone dedifferentiation. This includes their conclusions of 1) centrosome misorientation does not correlate with dedifferentiation. 2) dedifferentiated GSCs proliferate faster. These overstatements/ over-interpretations have to be corrected.

2) Dedifferentiation and re-differentiation are likely a dynamic process, and simple lineage tracing may not be sufficient to draw some conclusions described in this manuscript. Especially, the reviewer felt that it is not yet convincing to rely on lineage tracing (and counting clone number) to measure the activity of GSCs. It is equally conceivable that dedifferentiation causes fragmentation of the cysts, creating multiple 'clones' by fragmentation (thus, higher number of marked clones may not reflect higher proliferation rate). The authors would need additional evidence to make a point that dedifferentiated GSCs are more proliferative – ideally live imaging, or additional cell cycle markers such as mitotic index. If such experiments cannot be conducted to obtain conclusive results, the authors should not make such a statement.

3) In regards to the *puc>GFP* lineage tracing, the authors should present a time course to show whether labeled cells always originate from 4-cell and 8-cell cysts, as opposed to GSCs, GBs, and 2-cell cysts.

*Reviewer #1:*

In this manuscript, Herrera and Bach reports an exciting study that examines the biological significance of dedifferentiation in *Drosophila* testis. They used an elegant genetic technique to block dedifferentiation (however, note that this blockade of differentiation is not expected to be complete, see below) and found that dedifferentiation is required to maintain the GSC pool under 'challenging' conditions such as protein starvation. They further show that *JNK* pathway regulates dedifferentiation. The experiments are conducted to a high standard, supporting an exciting conclusion. I have several comments to be addressed prior to publication (they must be addressed, thus fall into 'major' comments, but they are quite straightforward to address).

- *bam-gal4* mediated experiments can only manipulate 4-cell spermatogonia and later, excluding the contribution of dedifferentiation from gonialblasts and 2-cell spermatogonia from testing (and those earlier SGs are considered to be more prone to dedifferentiation). Therefore, I suggest the authors to be cautious in stating 'dedifferentiation is not required under unchallenged conditions'. Their results demonstrate that the dedifferentiation from 4/8 SGs do not contribute to GSC maintenance during aging under 'normal' conditions. This clarification does not decrease the impact of this work, but I think it is important to mention throughout the text.

- Subsection “Dedifferentiation accelerates the recovery of the GSC pool after starvation”, last paragraph: the authors state that both native and dedifferentiated GSCs have equally high frequency of centrosome misorientation. Again this comes back to the point of 'GSCs dedifferentiated from gonialblast or 2-cell SGs'. Indeed, our prior work (Cheng et al., 2008) observed that centrosome misorientation increase during aging even in 'unmarked' GSCs (not marked by *bam-gal4* lineage tracing) and we speculated in this paper that this may reflect dedifferentiation from GB or 2 cell SGs (the centrosome misorientation is speculated to result from not having the original mother centrosome, which would be true with dedifferentiation from GB or 2 cell SGs). I think it is important to acknowledge the likely contribution of GB and 2-SG in dedifferentiation (especially because they are considered to be more prone to dedifferentiation compared to 4 or 8 SGs).

- In Figure 3 they show that presence of female impact dedifferentiation, and reveals the need of dedifferentiation (*bam>bam* flies decrease GSC pool). This is exciting (and I rather consider this as 'physiological condition' as flies in wild are supposed to be mating, than not mating at all in specific laboratory condition). I wonder if constantly supplying virgin females might further reveal the importance of dedifferentiation. They add virgin females in 'challenging' conditions (repeated cycle of starvation and mating etc. in Figure 4), but 'encountering virgin females periodically' should not be considered a challenge (it's physiological), and examining the impact of such a condition might reveal 'physiological role' of dedifferentiation.

- Figure 5: EdU incorporation cannot be used as a measure of cell cycle on its own, unless combined with mitotic index, because high EdU can simply mean that many cells are spending more time in S phase (instead of cycling faster).

- Discussion starts by stating 'dedifferentiation plays no role under standard aging conditions', which is an overstatement in my opinion because their experimental scheme cannot test the contribution of dedifferentiation from GB and 2 cell SG. They only discuss this possibility in the middle of Discussion, whereas they have never discussed this important possibility until this point. The authors should acknowledge this much earlier and fully integrate this possibility in the interpretation of their results starting in the Results section.

-Figure 4 title: typo, “comprised” should be compromised.

Reviewer #2:

This paper describes an investigation into the role of dedifferentiation in maintaining the GSC pool in *Drosophila* testes during chronically stressful conditions. Using a clever technique for genetically manipulating partially differentiated germ cells, the authors find that dedifferentiation of germ cells that have reached the *bam*(+) stage are important for replenishing the GSC pool following chronic starvation or continuous mating. They also show that dedifferentiated GSCs have increased rates of proliferation relative to GSCs that have not been derived from a *bam*(+) germ cell, and that *JNK* signaling is upregulated during chronic stress and required for efficient dedifferentiation. In general, I think their observations are novel and will be of broad interest. I also think that the data provide strong support for the central claims of the paper.

*Reviewer #3:*

In this manuscript, Herrera and Bach examine how the dedifferentiation of developing germ cells helps to maintain the germline stem cell pool in the *Drosophila* testis. The authors use a FLP-out lineage tracing system and over-express bam to block normal differentiation. They find dedifferentiation contributes little to the GSC pool in well-fed virgin males. By contrast, dedifferentiation does impact GSC numbers under "challenging" conditions, including mating and starvation. The authors provide evidence that GSCs arising from dedifferentiated cells are more active than their wild-type GSC siblings and that the *JNK* pathway is necessary for germ cell dedifferentiation in specific contexts.

The authors make several strong conclusions which are not necessarily supported by the data. In the Abstract, the authors write "Strikingly, blocking dedifferentiation under normal laboratory conditions has no impact on the GSC pool." Again, in the Results they state "Although unexpected and contrary to our predictions, these results preclude an important role for dedifferentiation of the *bam-Gal4* lineage in maintaining the GSC pool during aging under normal laboratory conditions." I think the authors should be much more explicit in the description of the conditions they are testing in both cases. They should clearly state that this observation is made in testes from unmated/virgin males, since I would consider mating a "normal" activity of male *Drosophila*. In regards to this experiment, the authors should also investigate or cite the relevant literature that describes how GSCs behave in unmated males. Are they as proliferative as those from mated males? Are they lost at the same rate (which should be assayed through clonal analysis)? If GSCs in unmated males are simply less active and are maintained for longer periods of time, the observed result is exactly what one would expect. Although the authors do mention the possibility in their discussion, making this statement in the Abstract ignores the possibility that two and four cell cysts (pre-*bam* expressing cells) dedifferentiate in unmated males. Examining centrosome orientation (which they use in later experiments) and live cell imaging could help clarify whether no dedifferentiation is occurring in these samples.

The authors draw the conclusion that dedifferentiation accelerates the recovery of the GSC pool after starvation. This conclusion is based on the observation that testes which have at least 1 *bam-GAL4* marked GSC "fully recovered" the pool of GSCs in 3 days, as opposed to those that have 0 dedifferentiated GSCs, which recovered the GSC pool in 7 days. This is a correlation, and the authors should characterize this phenotype in greater depth before making any firm conclusions. For example, the rate and levels of dedifferentiation may be exactly the same in both samples, but different testes may exhibit differences in cell death or GSC loss, in addition to any number of other variables, such as GSC proliferation and symmetric renewal. Again, live cell imaging may provide a more complete picture of how these cells are behaving in different circumstances.

The authors observe that dedifferentiated GSCs and their wild-type siblings display the same level of centrosome mis-orientation. This appears to be directly at odds with Cheng et al. Do the authors make this observation in all cases of dedifferentiation, or only in those testes from males subjected to starvation and refeeding? Further discussion of differences between this study and previous work is warranted.

The authors observe that dedifferentiated GSCs are more proliferative than their wild-type siblings based on the numbers of GFP positive cells within the testes. However, one cannot be certain that a particular labeled cell/cyst originated from a dedifferentiated GSC as opposed to a cyst that is in the process of breaking down. Dedifferentiation and redifferentiation may be a constant dynamic process. Do the authors have evidence that dedifferentiating cysts have to take on GSC identity before producing differentiating progeny? Again, the observation that dedifferentiated GSCs proliferate at a higher rate than wild-type cells contradicts previous studies. I would like to see live cell imaging that shows higher rates of cell division in the dedifferentiated GSCs and/or additional cell cycle/mitotic markers.

In regards to the *puc>GFP* lineage tracing, the authors should present a time course to show whether labeled cells always originate from 4-cell and 8-cell cysts, as opposed to GSCs, GBs, and 2-cell cysts?

[Editors' note: further revisions were requested prior to acceptance, as described below.]

Thank you for resubmitting your work entitled "*JNK* triggers dedifferentiation during chronic stress to maintain the germline stem cell pool in the *Drosophila* testis" for further consideration at *eLife*. Your revised article has been favorably evaluated by Marianne Bronner (Senior Editor), a Reviewing Editor, and two reviewers.

The manuscript has been improved but there are some remaining issues that need to be addressed before acceptance, as outlined below:

The reviewers appreciate that the authors addressed most of concerns raised during the first round of the review. Reviewer #3 raised a few points that other reviewers agreed to be important points. These points can be simply addressed by textual changes, and we would like the authors to edit the text to reflect on those points. After that, the reviewing editor should be able to editorially accept the manuscript.

*Reviewer #2:*

The authors have addressed my concerns and I think the paper is now ready for publication.

*Reviewer #3:*

The authors have addressed many of my previous concerns. I have just a couple of points that need further clarification.

The authors state: "Prior work has shown that the number of GSCs decreases slightly during aging under normal laboratory conditions, and this has led to the model that dedifferentiation provides a means to offset normal GSC loss during lifespan (Wang and Jones, 2011; Wallenfang, Nayak and DiNardo, 2006; Cheng et al., 2008). If this hypothesis is correct, we would expect a further reduction in GSC number after *bam* mis-expression by *bam-Gal4*.". They then present data that shows GSC numbers do not drop further when dedifferentiation is blocked, opposite from their expectations.

Previous results from the Matunis lab have shown that GSCs can undergo symmetric renewal. Perhaps blocking dedifferentiation forces the testis to use this alternative mechanism for maintaining GSC numbers. This does not necessarily mean that dedifferentiation does not play a role in maintaining GSC numbers in older flies. By blocking dedifferentiation, one may be shifting the equilibrium towards this and other potential mechanisms to maintain GSC numbers.

The authors go on to write "Although unexpected and contrary to our predictions, these results strongly suggest an important role for dedifferentiation of the *bam-Gal4* lineage in maintaining the GSC pool during aging under normal laboratory conditions." This conclusion seems at odds with the data that precedes it. Indeed, in the abstract the authors state "blocking *bam*-lineage dedifferentiation under normal conditions in virgin males has no impact on the GSC pool." Perhaps the authors could rephrase their concluding sentence in the Results section and mention the aforementioned alternative mechanisms.

The paper is entitled "*JNK* triggers dedifferentiation during chronic stress to maintain the germline stem cell pool in the *Drosophila* testis" The authors state in the final paragraph of the Results section- "To functionally test the relevance of *JNK* activity in spermatogonial dedifferentiation, we blocked its activity in the *bam-Gal4* lineage by mis-expressing the *JNK* inhibitor puc or a dominant negative form of the *JNK* basket (*bsk*). Concomitantly, we lineage traced *bam-Gal4* cells. If *JNK* activity is necessary for dedifferentiation, we predict that blocking its activity would reduce dedifferentiation upon refeeding. Indeed, after one cycle of starvation and refeeding, the increase in *bam*-lineage GSCs was no longer detectable in *bam>puc* or *bam>bsk^DN^* testes compared with *bam>LacZ* controls (Figure 6G-K). Taken together, these results indicate that *JNK* pathway activity is autonomously necessary for germline cells to undergo dedifferentiation."

Can the authors clarify whether cells expressing *puc* or *basketDN* survive and continue to differentiate? I do see *bam*-lineage positive cyst in Figure 6J, which partially speaks to this point. The worry is that over-expression of *puc* or *bsk^DN^* reduces germ cell viability, which could interfere with the interpretation of the data.

---

## [Author Response]

[…] The outline of essential revisions we ask is the following:1) There are strong concluding statements throughout the manuscript that are not necessarily backed up by the experimental evidence. (Also, there are several cases where the authors make rather strong statements in the Results, only to back off of them in the Discussion. These should be removed.)- There are several concerns regarding Bam-gal4-mediated lineage tracing of dedifferentiated GSCs. Although Bam-gal4-mediated lineage tracing indeed marks dedifferentiation, it does not mark all of dedifferentiated cells, because 2-cell SGs do not express bam-gal4. However, 2-cell SGs are considered to dedifferentiate more than 4- and 8-cell SGs due to their proximity to the niche. Thus, one would have to assume that there are many more dedifferentiated GSCs that are negative for lineage-marker than those positive for lineage-marker. Throughout the Results section, the authors do not sufficiently acknowledge this (highly likely) possibility, and draw strong conclusions, essentially assuming that all of linage-negative cells have never undergone dedifferentiation. This includes their conclusions of 1) centrosome misorientation does not correlate with dedifferentiation. 2) dedifferentiated GSCs proliferate faster. These overstatements/ over-interpretations have to be corrected.

We thank the reviewers for pointing this out. In the revised manuscript, we added several sentences/paragraphs to the Results and the Discussion that we are underlabeling dedifferentiated cells because it is believed that gonialblasts and 2-cell spermatogonia dedifferentiate but they are not labeled by *bam-Gal4*. We do not know the degree to which these events occur, but we now acknowledge that we are not monitoring all dedifferentiated cells. Furthermore, from Results onwards and in figures and legends, we specifically mention that we will refer to dedifferentiated cells as *bam*-lineage derived GSCs and the unlabeled GSCs as *bam*-lineage negative cells. We now discuss all conclusions in the context of *bam*-lineage cells.

We have also performed additional experiments to address the proportion of mis-oriented centrosomes in various conditions. At day 0, we find that *bam*-lineage GSCs have significantly more mis-oriented centrosomes than lineage-negative GSCs (see first set of bars in our Figure 2—figure supplement 1). This is in accordance with Cheng et al., 2008 (see their Figure 5E, second set of bars). At 45 days in fed, unmated, normal laboratory conditions, we find similar rates of mis-oriented centrosomes (~40-45%) in both *bam*-lineage positive and lineage-negative GSCs. This is quite similar to the 50 day time point reported by Cheng et al. (see their Figure 5E, last set of bars) where there was no significant difference between LacZ (i.e., *bam*-lineage)-positive and LacZ-negative GSCs. Additionally, we monitored centrosome mis-orientation after 4 cycles of chronic challenging conditions and found no significant difference in rates between *bam*-lineage positive and negative GSCs (see our Figure 2—figure supplement 1, last set of bars). These new data have been added to the revised manuscript (in Figure 2—figure supplement 1) and to the sixth paragraph of the Discussion. [In the original manuscript, we found no difference in centrosome mis-orientation in *bam*-lineage positive vs. negative GSCs after 1 cycle of starvation and refeeding, and these results are retained in the revised manuscript in Figure 2—figure supplement 1, third set of bars.] We have made two points regarding these results in the revised manuscript. First, we acknowledge that the lineage-negative GSCs likely include cells derived from gonialblasts and 2-cell gonia that are not label by *bam*-Gal4. We write: “It is possible that the lineage-negative GSCs are in fact dedifferentiated from gonialblasts and 2-cell gonia. While we cannot experimentally test this hypothesis, if both types of GSCs were in fact derived from dedifferentiated gonia, they would account the roughly equal rates of mis-oriented centrosomes in both pools.” Second, as there is no difference in centrosome mis-orientation between *bam*-lineage positive and negative GSCs in challenging conditions, we conclude that there is likely another explanation for the increased rate of proliferation of lineage-positive GSCs after 4 cycles. We write: “These data suggest that in some challenging conditions, factors independent of centrosome orientation may regulate GSC proliferation rates”.

As addressed in the next point, we found a higher rate of S- and M-phase in *bam*-lineage positive GSCs compared to lineage-negative ones. Therefore, even with the considerations described in the previous paragraph, lineage-positive GSCs do have a higher proliferation rate. This suggests that factors in addition to centrosome orientation regulate GSC proliferation. In the revised manuscript we speculate that lineage-positive GSCs have experienced autonomous *JNK* activity and this may “epigenetically reprogram” these cells, which may impact proliferation rates. In the revised manuscript, we carefully and precisely frame our results in the context of lineage (*bam* vs not *bam*) and still observe a significant increase in the rate of proliferation of lineage-positive GSCs.

2) Dedifferentiation and re-differentiation are likely a dynamic process, and simple lineage tracing may not be sufficient to draw some conclusions described in this manuscript. Especially, the reviewer felt that it is not yet convincing to rely on lineage tracing (and counting clone number) to measure the activity of GSCs. It is equally conceivable that dedifferentiation causes fragmentation of the cysts, creating multiple 'clones' by fragmentation (thus, higher number of marked clones may not reflect higher proliferation rate). The authors would need additional evidence to make a point that dedifferentiated GSCs are more proliferative – ideally live imaging, or additional cell cycle markers such as mitotic index. If such experiments cannot be conducted to obtain conclusive results, the authors should not make such a statement.

We agree with the reviewers’ concern about the need for additional evidence to support our conclusion that *bam*-lineage GSCs have a higher proliferation rate. To address this point, we subjected males to 4 cycles of challenging conditions and stained testes with anti-phospho Histone H3-Ser10 (pH3) (Millipore) that was recommended to us by experts in the field (Drs. Fuller, Yamashita and Jones). Using this antibody, we found that there were significantly more pH3-positive *bam*-lineage GSCs compared to lineage-negative GSCs (n=100 testes and >1,221 GSCs analyzed, p<0.053). These data have been added to the revised manuscript in Figure 5C and are discussed in the Results subsection “*bam*-lineage dedifferentiated GSCs are more proliferative than their siblings”. Taken together with the significant increase in S-phase in *bam*-lineage positive GSCs as well as the number of offspring (Figure 5A, B), we conclude that *bam*-lineage GSCs have a higher proliferation rate than lineage-negative GSCs. Given the ~2 month time frame for submitting a revision and the fact that we have not optimized technically-challenging, live imaging methodologies in our lab, we have not been able to perform live imaging experiments for this study.

3) In regards to the puc>GFP lineage tracing, the authors should present a time course to show whether labeled cells always originate from 4-cell and 8-cell cysts, as opposed to GSCs, GBs, and 2-cell cysts.

We thank the reviewers for pointing this out. To address this concern, we surveyed *puc>GFP* lineage labeling at two additional time points during refeeding (at 2 and 3 days). Based on prior work from the Matunis and Yamashita labs, we predicted that spermatogonia during the refeeding would experience transient *JNK* activity, fragment and migrate back to the niche. If this were the case, then we would expect to see only spermatogonia but not GSCs labeled at earliest time point. However, of the 97 testes examined, 17 (17.5%) had any germline labeling. Of these 17 testes, most (n=15) had at least 1 GSC labeled as well as various gonia. In fact, only 2 testes had gonia but no GSCs labeled. We speculate that the high number of cases where an entire germline lineage is labeled results from the considerable time (>24 hours) needed to induce *puc*, recombine the lineage-tracing cassette and then induce GFP. Nevertheless, our observations are consistent with *JNK*-activated gonia fragmenting and some of these gonial cells returning to the niche to revert to GSCs. We acknowledge that we cannot exclude the possibility that *JNK* signaling is directly activated in resident GSCs that have not dedifferentiated during the refeeding period. However, in support of our model that spermatogonia fragment and revert to the GSC state after autonomous *JNK* activation, we observed one testis at 3 days of refeeding where a 4-cell gonia labeled with *puc>GFP* appeared to fragment into a 3-cell gonia and a GSC at the niche (see new Figure 6—figure supplement 1). Within the 3-cell gonial cyst, each germ cell had a dot spectrosome, characteristic of fragmenting germ cells during dedifferentiation and very early germ cells in wild type testes. These data are discussed in the fourth paragraph of the subsection “The JNK pathway is necessary for *bam*-lineage spermatogonial dedifferentiation”.

Reviewer #1:[…] I have several comments to be addressed prior to publication (they must be addressed, thus fall into 'major' comments, but they are quite straightforward to address).- bam-gal4 mediated experiments can only manipulate 4-cell spermatogonia and later, excluding the contribution of dedifferentiation from gonialblasts and 2-cell spermatogonia from testing (and those earlier SGs are considered to be more prone to dedifferentiation). Therefore, I suggest the authors to be cautious in stating 'dedifferentiation is not required under unchallenged conditions'. Their results demonstrate that the dedifferentiation from 4/8 SGs do not contribute to GSC maintenance during aging under 'normal' conditions. This clarification does not decrease the impact of this work, but I think it is important to mention throughout the text.

We thank the reviewer for pointing this out. We acknowledge this and as discussed above in the response to reviewers, in the revised article we refer to dedifferentiated cells as *bam*-lineage derived and the other GSCs (which may be WT or dedifferentiated from gonialblasts or 2-cell spermatogonia) as lineage-negative.

- Subsection “Dedifferentiation accelerates the recovery of the GSC pool after starvation”, last paragraph: the authors state that both native and dedifferentiated GSCs have equally high frequency of centrosome misorientation. Again this comes back to the point of 'GSCs dedifferentiated from gonialblast or 2-cell SGs'. Indeed, our prior work (Cheng et al., 2008) observed that centrosome misorientation increase during aging even in 'unmarked' GSCs (not marked by bam-gal4 lineage tracing) and we speculated in this paper that this may reflect dedifferentiation from GB or 2 cell SGs (the centrosome misorientation is speculated to result from not having the original mother centrosome, which would be true with dedifferentiation from GB or 2 cell SGs). I think it is important to acknowledge the likely contribution of GB and 2-SG in dedifferentiation (especially because they are considered to be more prone to dedifferentiation compared to 4 or 8 SGs).

We agree that dedifferentiation of gonialblasts and 2-cell spermatogonia likely contribute to the pool of GSCs in testes from aged and challenged males. This is now acknowledged in numerous places in the text (see above).

- In Figure 3 they show that presence of female impact dedifferentiation, and reveals the need of dedifferentiation (bam>bam flies decrease GSC pool). This is exciting (and I rather consider this as 'physiological condition' as flies in wild are supposed to be mating, than not mating at all in specific laboratory condition). I wonder if constantly supplying virgin females might further reveal the importance of dedifferentiation. They add virgin females in 'challenging' conditions (repeated cycle of starvation and mating etc. in Figure 4), but 'encountering virgin females periodically' should not be considered a challenge (it's physiological), and examining the impact of such a condition might reveal 'physiological role' of dedifferentiation.

We agreed with this reviewer and with reviewer #3 that mating is a normal physiological response of males to the presence of females. However, mating is also an energy-intensive activity that can negatively impact males. For example, mated males have significantly shorter lifespans than unmated males (Branco, 2017). Hence, we conclude that mating, while normal, is stressful to males. In the text of the revised manuscript, we acknowledge both of these facts. Additionally, in response to reviewer #3 (see below), we have found that mating provokes an overall increase in GSC proliferation compared to aged-matched, unmated males (see Figure 3—figure supplement 1). This suggests that mating induces a systemic response that increases GSC division rates. Furthermore, again as described below, we find that neutral GSC clones are lost significantly from the testis niche upon mating, presumably due to increased proliferation rates of GSCs as well as outcompetition of resident GSCs by dedifferentiating germ cells.

- Figure 5: EdU incorporation cannot be used as a measure of cell cycle on its own, unless combined with mitotic index, because high EdU can simply mean that many cells are spending more time in S phase (instead of cycling faster).

We agree with the reviewer. To address this point, we stained testes from control *bam>LacZ* males after 4 cycles of challenging conditions with the anti-pH3 described above. We found significantly more *bam*-lineage positive GSCs in M-phase compared to lineage-negative GSCs (p<0.053). These raw data are in the new Supplementary file 2 and also presented in Figure 5C in the revised manuscript. Taken together with our prior EdU incorporation results (which was significantly higher in *bam*-lineage GSCs), we conclude that *bam*-lineage GSCs cycle faster than lineage-negative GSCs.

- Discussion starts by stating 'dedifferentiation plays no role under standard aging conditions', which is an overstatement in my opinion because their experimental scheme cannot test the contribution of dedifferentiation from GB and 2 cell SG. They only discuss this possibility in the middle of Discussion, whereas they have never discussed this important possibility until this point. The authors should acknowledge this much earlier and fully integrate this possibility in the interpretation of their results starting in the Results section.

We thank the reviewer for point this out. As discussed in detail above, we have now incorporated this possibility into the first paragraph of the Results where we describe the lineage tracing. We now treat in greater detail this possibility in the Discussion.

-Figure 4 title: typo, “comprised” should be compromised.

Thank you for pointing out this typo. We have fixed this in the revised manuscript.

Reviewer #3:[…] The authors make several strong conclusions which are not necessarily supported by the data. In the Abstract, the authors write "Strikingly, blocking dedifferentiation under normal laboratory conditions has no impact on the GSC pool." Again, in the Results they state "Although unexpected and contrary to our predictions, these results preclude an important role for dedifferentiation of the bam-Gal4 lineage in maintaining the GSC pool during aging under normal laboratory conditions." I think the authors should be much more explicit in the description of the conditions they are testing in both cases. They should clearly state that this observation is made in testes from unmated/virgin males, since I would consider mating a "normal" activity of male Drosophila.

We thank the reviewer for pointing this out. As described above, we now clearly indicate that these initial experiments were performed on unmated males. We have also updated the Abstract to reflect the fact that *bam*-lineage dedifferentiation does not impact the GSC pool in unmated males but does play an important role in maintaining this pool of stem cells in mated males and/or stressful conditions (like starvation).

In regards to this experiment, the authors should also investigate or cite the relevant literature that describes how GSCs behave in unmated males. Are they as proliferative as those from mated males? Are they lost at the same rate (which should be assayed through clonal analysis)? If GSCs in unmated males are simply less active and are maintained for longer periods of time, the observed result is exactly what one would expect. Although the authors do mention the possibility in their discussion, making this statement in the Abstract ignores the possibility that two and four cell cysts (pre-bam expressing cells) dedifferentiate in unmated males. Examining centrosome orientation (which they use in later experiments) and live cell imaging could help clarify whether no dedifferentiation is occurring in these samples.

The reviewer has raised some important concerns. First, we were unable to find any literature on the effects of mating on GSC proliferation. As such, we performed experiments to directly address this. We show in Figure 3—figure supplement 1 of the revised manuscript that the rate of mitosis (pH3-positive GSCs) is significantly higher in testes from mated males compared to age-matched, unmated males. Moreover, we analyzed the mitotic rate of *bam*-lineage-positive GSCs vs. lineage-negative GSCs in both conditions. We found that both types of GSCs have significantly higher mitotic rates compared to those in unmated controls (6.44% vs. 9.56% in lineage-negative cells (p<0.068) and 6.15% vs 13.61% in lineage-positive cells (p<0.034)). These data suggest that there is a systemic response to mating that increases proliferation of all GSCs.

We also performed the clonal analysis suggested by the reviewer. We made neutral *FRT^40A^* MARCM clones with which we have considerable experience (1, 2) in unmated vs. mated males at 2 and 15 days post clone induction (dpci). Both types of males had similar clone induction at 2 dpci, as expected. However, there were significantly fewer GSC clones at 15 dpci in the testes of mated males compared to unmated males. The GSC clone loss in mated flies is compatible with the increased rate of GSC proliferation discussed in the previous paragraph. Furthermore, it is also consistent with the presumed competition of dedifferentiating germ cells with resident GSCs for niche occupancy. These data are included in new Figure 3—figure supplement 1 and are discussed in the Results subsection "*bam*-lineage dedifferentiation preserves both the GSC pool and spermatogenesis during chronic challenging conditions”.

The authors draw the conclusion that dedifferentiation accelerates the recovery of the GSC pool after starvation. This conclusion is based on the observation that testes which have at least 1 bam-GAL4 marked GSC "fully recovered" the pool of GSCs in 3 days, as opposed to those that have 0 dedifferentiated GSCs, which recovered the GSC pool in 7 days. This is a correlation, and the authors should characterize this phenotype in greater depth before making any firm conclusions. For example, the rate and levels of dedifferentiation may be exactly the same in both samples, but different testes may exhibit differences in cell death or GSC loss, in addition to any number of other variables, such as GSC proliferation and symmetric renewal. Again, live cell imaging may provide a more complete picture of how these cells are behaving in different circumstances.

We concur with the reviewer and have now moderated our conclusions about the correlation between the number of *bam*-lineage-positive GSCs and the recovery of the GSC pool. We now write: “Although we cannot exclude other variables such as germ cell death, GSC loss, and GSC gain through symmetric renewal, this correlation strongly suggests that *bam*-lineage dedifferentiation accelerates the recovery of the GSC pool under challenging conditions.” As noted above, given the time frame for revision and not having optimized live imaging methodology in our lab, which is technically challenging, we have not performed live imaging experiments for this study.

The authors observe that dedifferentiated GSCs and their wild-type siblings display the same level of centrosome mis-orientation. This appears to be directly at odds with Cheng et al. Do the authors make this observation in all cases of dedifferentiation, or only in those testes from males subjected to starvation and refeeding? Further discussion of differences between this study and previous work is warranted.

In the original submission, we had examined centrosome orientation in GSCs after 15 days of starvation and 5 days of refeeding. Based on this reviewer’s concern, we examined centrosome orientation in several other conditions, specifically, 0 days, after 45 days in fed but unmated males, and after 4 cycles of starvation, refeeding and mating. We already provided a detailed response to this concern above in our answer to the first general point (see above).

The authors observe that dedifferentiated GSCs are more proliferative than their wild-type siblings based on the numbers of GFP positive cells within the testes. However, one cannot be certain that a particular labeled cell/cyst originated from a dedifferentiated GSC as opposed to a cyst that is in the process of breaking down. Dedifferentiation and redifferentiation may be a constant dynamic process. Do the authors have evidence that dedifferentiating cysts have to take on GSC identity before producing differentiating progeny? Again, the observation that dedifferentiated GSCs proliferate at a higher rate than wild-type cells contradicts previous studies. I would like to see live cell imaging that shows higher rates of cell division in the dedifferentiated GSCs and/or additional cell cycle/mitotic markers.

We agree with the reviewer and as described above we measured mitotic rate with pH3. This experiment revealed that there were significantly more *bam*-lineage-positive GSCs labeled with pH3 compared to lineage-negative GSCs (see above). Regarding dedifferentiation and redifferentiation, we have not been able to perform live imaging studies that may reveal such characteristics. However, in response to reviewer 3’s next comment (also addressed above in response to general question 3), we have now analyzed a time course of *puc*-lineage activation during the refeeding period. Please see above for a detailed response, but suffice it to say here, we find some evidence for fragmentation of 4-cell gonia and migration of one or more germ cells from this cyst back to the niche to re-acquire GSC status.

In regards to the puc>GFP lineage tracing, the authors should present a time course to show whether labeled cells always originate from 4-cell and 8-cell cysts, as opposed to GSCs, GBs, and 2-cell cysts?

This is an excellent suggestion. Please see above answer to general question 3.

Literature cited:

1) Amoyel M, Anderson J, Suisse A, Glasner J, Bach EA. Socs36E Controls Niche Competition by Repressing MAPK Signaling in the *Drosophila* Testis. PLoS Genet. 2016;12(1):e1005815. 10.1371/journal.pgen.1005815

2) Amoyel M, Simons BD, Bach EA. Neutral competition of stem cells is skewed by proliferative changes downstream of Hh and Hpo. EMBO J. 2014;33(20):2295-313. 10.15252/embj.201387500

[Editors' note: further revisions were requested prior to acceptance, as described below.]

The manuscript has been improved but there are some remaining issues that need to be addressed before acceptance, as outlined below:The reviewers appreciate that the authors addressed most of concerns raised during the first round of the review. Reviewer #3 raised a few points that other reviewers agreed to be important points. These points can be simply addressed by textual changes, and we would like the authors to edit the text to reflect on those points. After that, the reviewing editor should be able to editorially accept the manuscript.Reviewer #3:The authors have addressed many of my previous concerns. I have just a couple of points that need further clarification.The authors state: "Prior work has shown that the number of GSCs decreases slightly during aging under normal laboratory conditions, and this has led to the model that dedifferentiation provides a means to offset normal GSC loss during lifespan (Wang and Jones, 2011; Wallenfang, Nayak and DiNardo, 2006; Cheng et al., 2008). If this hypothesis is correct, we would expect a further reduction in GSC number after bam mis-expression by bam-Gal4.". They then present data that shows GSC numbers do not drop further when dedifferentiation is blocked, opposite from their expectations.Previous results from the Matunis lab have shown that GSCs can undergo symmetric renewal. Perhaps blocking dedifferentiation forces the testis to use this alternative mechanism for maintaining GSC numbers. This does not necessarily mean that dedifferentiation does not play a role in maintaining GSC numbers in older flies. By blocking dedifferentiation, one may be shifting the equilibrium towards this and other potential mechanisms to maintain GSC numbers.

We thank reviewer 3 for pointing this out. In this second revision we now directly acknowledge this possible shift in equilibrium towards increased symmetric renewal when *bam*-lineage dedifferentiation is inhibited. We write “Additionally, prior work has shown that symmetric renewal, whereby a gonialblast swivels to gain direct access to the niche, occurs at low levels in testes from wild type males (Sheng and Matunis, 2011). It is possible that by blocking *bam*-lineage dedifferentiation, we are shifting the equilibrium towards symmetric renewal, but live imaging will be needed to test this hypothesis.”

The authors go on to write "Although unexpected and contrary to our predictions, these results strongly suggest an important role for dedifferentiation of the bam-Gal4 lineage in maintaining the GSC pool during aging under normal laboratory conditions." This conclusion seems at odds with the data that precedes it. Indeed, in the abstract the authors state "blocking bam-lineage dedifferentiation under normal conditions in virgin males has no impact on the GSC pool." Perhaps the authors could rephrase their concluding sentence in the Results section and mention the aforementioned alternative mechanisms.

We are extremely grateful to reviewer 3 for pointing this out. In fact, we made a terrible omission in this sentence in the original revision. We had intended to state that *bam*-lineage dedifferentiation does *not* play a critical role in maintaining the GSC pool under normal laboratory conditions. However, we forgot the negative. In the revised manuscript we now clearly state our original intention. “Nevertheless, although unexpected and contrary to our predictions, our results strongly suggest that dedifferentiation of the *bam*-lineage does not play an important role in maintaining the GSC pool during aging under normal laboratory conditions.”

The paper is entitled "JNK triggers dedifferentiation during chronic stress to maintain the germline stem cell pool in the Drosophila testis" The authors state in the final paragraph of the Results section- "To functionally test the relevance of JNK activity in spermatogonial dedifferentiation, we blocked its activity in the bam-Gal4 lineage by mis-expressing the JNK inhibitor puc or a dominant negative form of the JNK basket (bsk). Concomitantly, we lineage traced bam-Gal4 cells. If JNK activity is necessary for dedifferentiation, we predict that blocking its activity would reduce dedifferentiation upon refeeding. Indeed, after one cycle of starvation and refeeding, the increase in bam-lineage GSCs was no longer detectable in bam>puc or bam>bsk^DN^ testes compared with bam>LacZ controls (Figure 6G-K). Taken together, these results indicate that JNK pathway activity is autonomously necessary for germline cells to undergo dedifferentiation."Can the authors clarify whether cells expressing puc or basketDN survive and continue to differentiate? I do see bam-lineage positive cyst in Figure 6J, which partially speaks to this point. The worry is that over-expression of puc or bsk^DN^ reduces germ cell viability, which could interfere with the interpretation of the data.

Reviewer 3 has raised an important point, that inhibition of *JNK* may be negatively impacting the health of germ cells. To address this concern, we have examined at least 15 testes of the *bam>puc* or *bam>bsk^DN^* genotypes. In all cases, we found evidence for robust germ cell differentiation into pre-meiotic stages. Representative examples are included in a new supplementary figure (Figure 6—figure supplement 2). Additionally, we write “We note that neither mis-expression of *puc* nor *bsk^DN^* appears to adversely affect germ cell differentiation as robust spermatogonia and spermatocytes expressing *puc* or *bsk^DN^* survive and continue to progress towards meiotic stages (Figure 6—figure supplement 2).” We have also provided a legend for the new figure supplement.